EMBO
Molecular Medicine

# Dual inhibition of AKT-mTOR and AR signaling by targeting HDAC3 in *PTEN*- or *SPOP*-mutated prostate cancer

Yuqian Yan[1,2,†], Jian An[2,†,‡], Yinhui Yang[2,3,†], Di Wu[2], Yang Bai[2,3], William Cao[2], Linlin Ma[2,4], Junhui Chen[5], Zhendong Yu[6], Yundong He[2], Xin Jin[2], Yunqian Pan[2], Tao Ma[7], Shangqian Wang[8], Xiaonan Hou[9], Saravut John Weroha[9], R Jeffrey Karnes[10], Jun Zhang[11], Jennifer J Westendorf[2], Liguo Wang[7], Yu Chen[8], Wanhai Xu[3], Runzhi Zhu[4,*] [iD], Dejie Wang[1,2,**] [iD] & Haojie Huang[2,10,12,***] [iD]

## Abstract

AKT-mTOR and androgen receptor (AR) signaling pathways are aberrantly activated in prostate cancer due to frequent PTEN deletions or SPOP mutations. A clinical barrier is that targeting one of them often activates the other. Here, we demonstrate that HDAC3 augments AKT phosphorylation in prostate cancer cells and its overexpression correlates with AKT phosphorylation in patient samples. HDAC3 facilitates lysine-63-chain polyubiquitination and phosphorylation of AKT, and this effect is mediated by AKT deacetylation at lysine 14 and 20 residues and HDAC3 interaction with the scaffold protein APPL1. Conditional homozygous deletion of *Hdac3* suppresses prostate tumorigenesis and progression by concomitant blockade of AKT and AR signaling in the *Pten* knock-out mouse model. Pharmacological inhibition of HDAC3 using a selective HDAC3 inhibitor RGFP966 inhibits growth of both PTEN-deficient and SPOP-mutated prostate cancer cells in culture, patient-derived organoids and xenografts in mice. Our study identifies HDAC3 as a common upstream activator of AKT and AR signaling and reveals that dual inhibition of AKT and AR pathways is achievable by single-agent targeting of HDAC3 in prostate cancer.

**Keywords** AKT phosphorylation; androgen receptor; HDAC3; prostate cancer; RGFP966

**Subject Categories** Cancer; Pharmacology & Drug Discovery

See also: **A Zoubeidi & ME Gleave** (April 2018)

## Introduction

The majority of prostate cancers are dependent on androgens and activation of the androgen receptor (AR) for growth and survival, and androgen deprivation therapy remains the mainstay of treatment for advanced prostate cancer (Watson *et al*, 2015). The PI3K/AKT pathway is often aberrantly activated due to frequent deletion or mutation of the *PTEN* tumor suppressor gene and activation mutations in *PIK3CA* and *AKT1* genes during prostate tumorigenesis and progression (Cancer Genome Atlas Research Network, 2015, Robinson *et al*, 2015), representing another key actionable target. However, both patient data analysis and pre-clinical animal model studies invariably show that loss of PTEN promotes enhanced AKT activity and reduced AR signaling and that inhibition of AKT results in AR activation while blockade of AR function increases AKT activities (Carver *et al*, 2011; Mulholland *et al*, 2011), stressing the

1  Department of Gastroenterology, Jiangxi Institute of Gastroenterology and Hepatology, First Affiliated Hospital of Nanchang University, Nanchang, Jiangxi, China
2  Department of Biochemistry and Molecular Biology, Mayo Clinic College of Medicine, Rochester, MN, USA
3  Department of Urology, The Fourth Hospital of Harbin Medical University, Harbin, Heilongjiang, China
4  Center for Cell Therapy, The Affiliated Hospital of Jiangsu University, Zhenjiang, Jiangsu, China
5  Department of Minimally Invasive Intervention, Peking University Shenzhen Hospital, Shenzhen, Guangdong, China
6  Central Laboratory, Peking University Shenzhen Hospital, Shenzhen, Guangdong, China
7  Department of Biomedical Statistics and Informatics, Mayo Clinic Cancer Center, Rochester, MN, USA
8  Human Oncology and Pathogenesis Program, Memorial Sloan-Kettering Cancer Center, New York, NY, USA
9  Department of Oncology, Mayo Clinic College of Medicine, Rochester, MN, USA
10 Department of Urology, Mayo Clinic College of Medicine, Rochester, MN, USA
11 Department of Laboratory Medicine and Pathology, Mayo Clinic College of Medicine, Rochester, MN, USA
12 Mayo Clinic Cancer Center, Mayo Clinic College of Medicine, Rochester, MN, USA
   *Corresponding author. Tel: +86 511-84405370; E-mail: runzhizhu1978@163.com
   **Corresponding author. Tel: +86 79188692507, E-mail: wdj5257@sina.com
   ***Corresponding author. Tel: +1 507-293-1712, E-mail: huang.haojie@mayo.edu
   †These authors contributed equally to this work
   ‡Present address: Dana-Farber Cancer Institute, Harvard Medical School, Boston, MA, USA

requirement of co-targeting of both pathways for effective treatment of prostate cancer.

HDACs belong to a super family of proteins. In humans, there are 18 HDAC proteins categorized into four distinct classes (I, II, III, and IV) according to their homology to yeast proteins, subcellular location, and enzymatic activities (de Ruijter et al, 2003; Gallinari et al, 2007). The HDAC family was originally found to be involved in deacetylating the histone core of nucleosomes to configure chromosomal structure and regulate gene expression (Taunton et al, 1996). Also, it is implicated in regulating deacetylation and phosphorylation of non-histone proteins (Kouzarides, 2000; Kramer et al, 2009). In fact, HDAC inhibitors (HDACIs) have been developed for cancer therapy (Dokmanovic et al, 2007), supporting the critical oncogenic role of HDACs in tumorigenesis.

HDAC3, a class I HDAC, plays critical roles in S phase progression, DNA damage control, maintenance of genomic stability, and T-cell development (Bhaskara et al, 2008, 2010; Wang et al, 2015). HDAC3 is overexpressed in a majority of prostate cancers (Weichert et al, 2008), implying a role of HDAC3 in prostate tumorigenesis. Depletion of HDAC3 or other HDACs suppresses expression of AR and its downstream target genes in prostate cancer cells, although the underlying mechanism remains poorly understood (Welsbie et al, 2009). Additionally, it has been shown that HDAC3 knockdown dramatically reduces leukemia and lymphoma cell proliferation (Matthews et al, 2015). These data suggest that HDAC3 could be a therapeutic target for cancers such as those in the prostate.

Speckle-type POZ protein (SPOP) is the substrate-binding adaptor of the CULLIN3-RBX1 E3 ubiquitin ligase complex (Zhuang et al, 2009). The gene encoding SPOP is the most frequently mutated gene in human primary prostate cancers (Barbieri et al, 2012; Cancer Genome Atlas Research Network, 2015). Functional studies show that ectopic expression of the most frequent SPOP-mutant F133V in human prostatic cells or knock-in in the mouse prostate results in aberrant activation of AR and AKT-mTORC1 signaling (An et al, 2014, 2015; Geng et al, 2014; Blattner et al, 2017; Zhang et al, 2017). Most importantly, the Cancer Genome Atlas (TCGA) data demonstrate that SPOP-mutated prostate cancers exhibit the highest AR activity among all molecular subtypes of prostate cancer examined (Cancer Genome Atlas Research Network, 2015). Thus, it is important to identify a common target to inhibit both AKT-mTORC1 and AR signaling in SPOP-mutated prostate cancer.

In the present study, we demonstrated that HDAC3 is required for AKT phosphorylation in prostate cancer cells. Prostate-specific knockout of Hdac3 decreased Akt phosphorylation, alleviated the tumor burden, and ultimately prolonged survival of Pten knockout mice. In human prostate cancer organoids and xenograft models, we further showed that a selective HDAC3 inhibitor is efficacious in inhibition of AKT and AR signaling in both PTEN- and SPOP-mutant background.

## Results

### HDAC3 is the only class I/II HDAC protein that regulates AKT phosphorylation

It has been shown previously that different pan class I/II HDACIs have differential effects on AKT phosphorylation at both threonine

308 (T308) and serine 473 (S473) in AR-negative prostate cancer PC-3 cells (Chen et al, 2005a). By treating AR-positive prostate cancer C4-2 cells with the commonly used pan class I/II HDACIs trichostatin A (TSA), suberoylanilide hydroxamic acid (SAHA), panobinostat (LBH589), and HDAC6-selective inhibitor tubastatin A, we demonstrated that these pan class I/II HDACIs, but not tubastatin A completely inhibited AKT phosphorylation at S473 and T308 (Fig 1A). To further clarify whether this was due to the possibility that these HDACIs potentially regulate the expression of upstream regulators (e.g., CXCR7 or PHLPP1) (Wang et al, 2008; Luan et al, 2009; Bradley et al, 2013), and thereby indirectly affect AKT phosphorylation, we treated C4-2 cells with cycloheximide (CHX) to block de novo protein synthesis. To our surprise, CHX treatment only had very minimal effect on pan HDACI-induced inhibition of AKT phosphorylation (Fig 1A), suggesting that decreased AKT phosphorylation by pan class I/II HDACIs was not primarily mediated by their effect on expression of AKT upstream regulators.

To identify which member(s) in the class I/II HDAC family is the major modulator of AKT phosphorylation, all 11 members in these subfamilies were knocked down individually by two independent small hairpin RNAs (shRNAs). Each HDAC gene was effectively knocked down to 40% or more at mRNA level (Fig 1B). Notably, only HDAC3 knockdown substantially decreased AKT phosphorylation at both T308 and S473 residues in a similar degree in C4-2 cells (Fig 1C). We also examined the effect of a selective HDAC3 inhibitor, RGFP966, on AKT phosphorylation. We demonstrated that RGFP966 inhibited AKT phosphorylation as early as 0.5 h post-treatment (Fig 1D), further suggesting a direct effect of HDAC3 inhibition of AKT phosphorylation. Together, these data indicate that HDAC3, but not other class I/II HDACs, is primarily required for AKT phosphorylation in this cell line.

Consistent with a previous finding that class I HDAC members (HDAC1, 2, 3, and 8) are highly expressed in prostate cancers (Weichert et al, 2008), analysis of TCGA data also showed that expression of these four HDAC genes was upregulated at the mRNA level in tumors compared to normal tissues (Fig EV1A). Specifically, comparison of 52 paired normal and tumor samples showed that the majority of them [approximately 56% (29 out of 52)] exhibited an increased expression of HDAC3 at the mRNA level in tumors (Fig EV1B), suggesting that HDAC3 is a highly relevant protein in prostate cancer. We further examined the correlation between HDAC3 protein expression and AKT phosphorylation by performing immunohistochemistry (IHC) on a tissue microarray (TMA) containing 55 prostate cancer samples. We demonstrated that increased expression of HDAC3 correlated with higher levels of AKT phosphorylation (S473) in this cohort of patients (Fig 1E and F). Therefore, HDAC3 might be an essential upstream regulator of AKT phosphorylation in prostate cancer cells in culture and in patients.

### HDAC3 is required for growth factor-induced AKT polyubiquitination and activation

Polyubiquitination is a critical step for growth factor-induced phosphorylation and activation of AKT (Yang et al, 2009). Given that acetylation and polyubiquitination can compete with each other by occurring at the same lysine residues (Yang & Seto, 2008), we investigated whether HDAC3 regulates AKT acetylation and polyubiquitination. Firstly, we showed that HDAC3 overexpression substantially

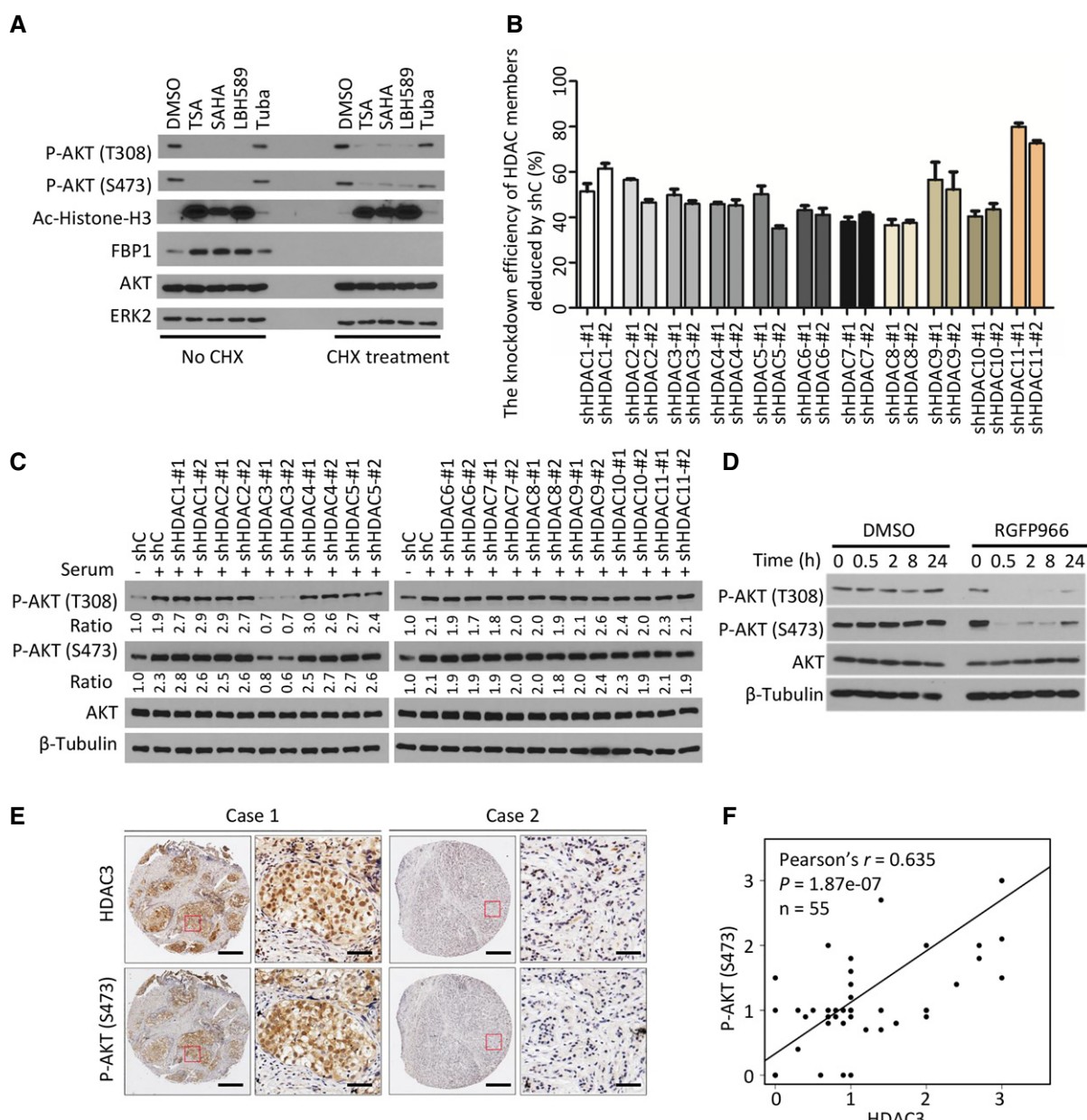

**Figure 1.  HDAC3 regulates AKT phosphorylation.**

A   HDACIs inhibited AKT phosphorylation. C4-2 cells were pre-treated with 20 μM of CHX for 30 min followed by treatment with pan HDACIs TSA (1 μM), SAHA (5 μM), LBH589 (0.1 μM), or a HDAC6 selective inhibitor Tuba (5 μM) for 24 h prior to Western blot analysis with indicated antibodies. The efficacy of CHX was evident by blockade of induction of FBP1 expression by HDACIs as reported (Yang *et al*, 2017).

B   The knockdown efficiency of each member of class I/II HDACs by shRNAs. C4-2 cells were stably infected with control or gene-specific shRNAs for 4 days and harvested for RT–qPCR. Expression of each gene was first normalized to the level of *GAPDH*, and then the expression level of each gene in gene knockdown cells was normalized by that in control knockdown cells. The shRNA knockdown efficiency was determined by subtracting the normalized value from 100%. Data represents means ± SEM. The RT–qPCR was performed in triplicate for each sample.

C   Control or gene-specific knockdown C4-2 cells were serum starved for 24 h and then cultured in regular culture medium for 12 h followed by Western blots for indicated proteins. Western blot bands for total and phosphorylated AKT were quantified and normalized to the quantified values of β-Tubulin (loading control). The normalized values were further normalized to the value of shC-infected cells without serum stimulation.

D   C4-2 cells were treated with vehicle (DMSO) or HDAC3 inhibitor RGFP966, and at different time points, cells were harvested for Western blots with the indicated antibodies.

E   The representatives of IHC staining for HDAC3 and AKT S473 phosphorylation in prostate cancer patient specimens; scale bar: 50 μm; scale bar for the inset: 20 μm.

F   Correlation between expression of AKT S473 phosphorylation and HDAC3 was shown, *n* = 55, ***P = 1.87e-07 was performed by Pearson's product-moment correlation test.

Source data are available online for this figure.

decreased AKT acetylation without affecting total AKT level in C4-2 cells (Fig 2A). Decreased AKT acetylation was accompanied with increased polyubiquitination of AKT, and the effect was dose-dependent (Fig 2B). In agreement with the observation that HDAC3 knockdown decreased AKT phosphorylation (Fig 1C), overexpression of HDAC3 increased AKT phosphorylation at both T308 and S473 sites (Fig 2C). In contrast, depletion of HDAC3 by a pool of three independent siRNAs increased AKT acetylation (Fig 2D), but decreased AKT polyubiquitination and phosphorylation (Fig 2D and E). The HDAC3 inhibitor RGFP966 also undermined AKT ubiquitination (Fig 2F).

Growth factors, such as insulin-like growth factor-1 (IGF-1) and epidermal growth factor (EGF), are the potent upstream stimulators of AKT signaling pathway (Song et al, 2005; Morgan et al, 2009).

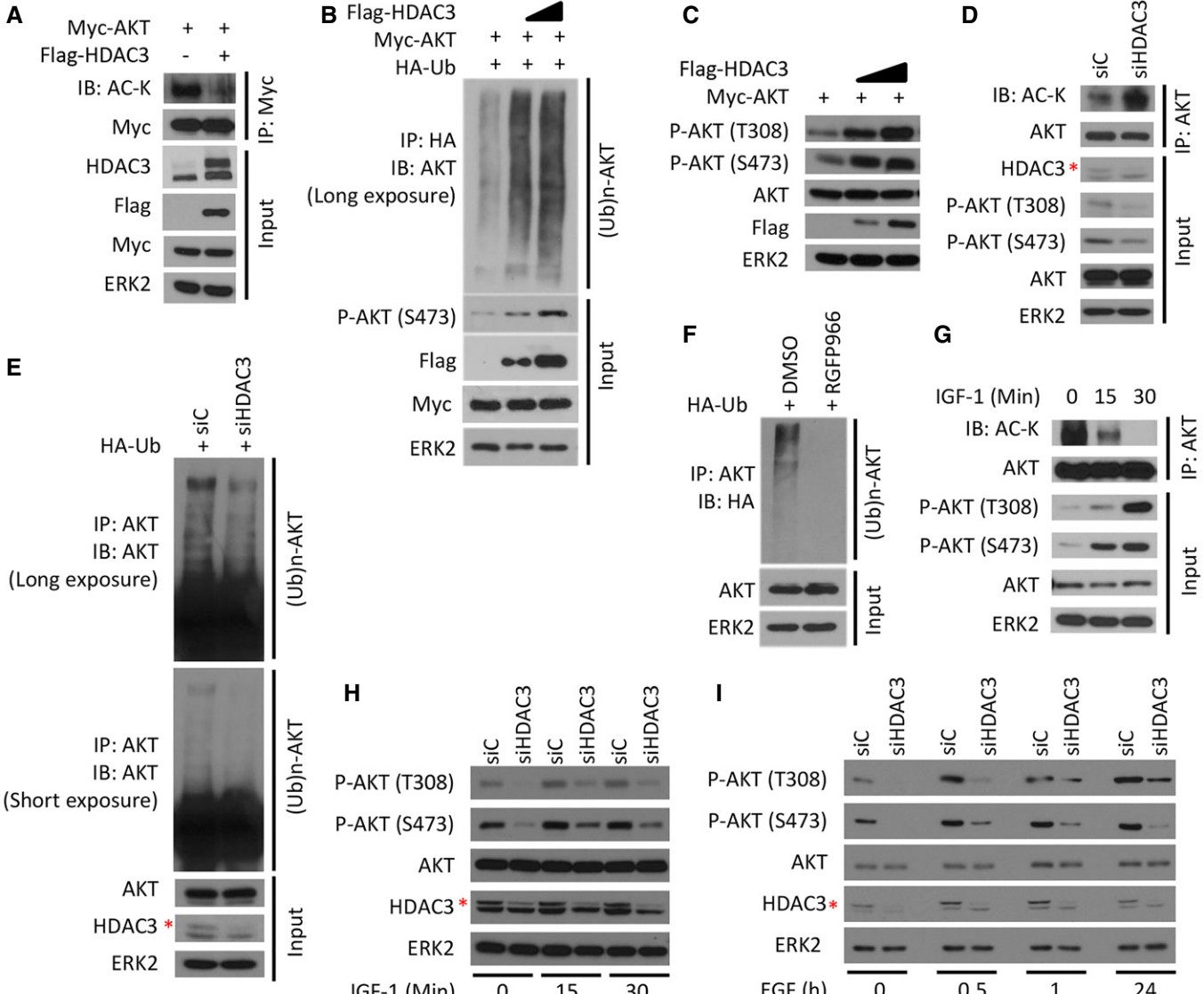

**Figure 2.  HDAC3 is important for growth factor-induced AKT deacetylation and polyubiquitination.**

A    293T cells were transfected with plasmids for Flag-HDAC3 and Myc-AKT followed by IP and Western blots with the indicated antibodies.

B    C4-2 cells were transfected with plasmids for Myc-AKT, HA-Ub, and Flag-HDAC3 (different doses) for 24 h followed by IP and Western blots with the indicated antibodies.

C    C4-2 cells were transfected with plasmids for Myc-AKT and Flag-HDAC3 (different doses) for 24 h followed by Western blots with the indicated antibodies.

D, E    C4-2 cells were transfected with a pool of control or HDAC3-specific siRNAs for 48 h followed by IP and/or Western blots with the indicated antibodies.

F    C4-2 cells were transfected with HA-Ub plasmids and treated with 3 µM of RGFP966 for 24 h followed by IP and/or Western blots with the indicated antibodies.

G    C4-2 cells were treated with 10 ng/ml of IGF-1 for different periods of time and harvested for IP and Western blots with the indicated antibodies.

H, I    C4-2 cells were transfected with a pool of control and HDAC3-specific siRNAs for 48 h and then treated with IGF-1 (H) or EGF (I) followed by Western blots for indicated proteins.

Data information: The asterisk (*) indicates the specific HDAC3 protein band.
Source data are available online for this figure.

To examine whether growth factor stimulation affects AKT acetylation, we treated C4-2 cells with IGF-1 and found that AKT acetylation was inhibited by IGF-1 in a dose-dependent manner, while its phosphorylation at both T308 and S473 was concomitantly upregulated (Fig 2G). These data further confirmed that AKT phosphorylation is negatively regulated by its acetylation. Most importantly, knockdown of HDAC3 abolished IGF-1-induced elevation of AKT phosphorylation (Fig 2H). Similar to IGF-1, EGF is another well-known growth factor that activates the PI3K/AKT signaling pathway. Similarly, we found that AKT phosphorylation at T308 and S473 sites was enhanced by EGF treatment in a time-dependent manner, but the magnitude of AKT activation was substantially diminished in HDAC3 knockdown cells (Fig 2I). Our data show that HDAC3 is required for growth factor-stimulated phosphorylation of AKT.

### A functional AKT-binding domain is identified in HDAC3

To understand the molecular mechanisms by which HDAC3 regulates acetylation, polyubiquitination, and phosphorylation of AKT, we examined whether HDAC3 interacts with AKT. Reciprocal co-immunoprecipitation (co-IP) assays showed that both ectopically expressed and endogenous HDAC3 and AKT proteins were detectable in the same protein complex in 293T and C4-2 cell lines, respectively (Fig 3A–C). In contrast, consistent with the unbiased screening results (Fig 1B and C), endogenous AKT did not interact with endogenous HDAC1 and HDAC2 in C4-2 cells (Fig 3D).

To identify the specific region of HDAC3 that is essential for AKT binding, we generated four glutathione-S-transferase (GST)-HDAC3 recombinant proteins (P1, P2, P3, and P4; Fig 3E). GST pull-down assays showed that GST-HDAC3 P2 (amino acids (aa) 101–200), but not GST or other GST-HDAC3 recombinant proteins, specifically interacted with Myc-tagged AKT (Fig 3F). To further narrow down the specific fragment for AKT binding, we constructed five additional GST-HDAC3 recombinant proteins (HDAC3-Δ1, 2, 3, 4, and 5) by sequentially deleting 20 amino acids from GST-HDAC3 P2 recombinant protein (aa 101–120, 121–140, 141–160, 161–180, and 181–200). GST pull-down assay showed that loss of aa 141–160 in HDAC3 completely abolished its interaction with AKT (Fig 3G), and we termed this region as AKT-binding domain (ABD). Unlike the wild-type HDAC3 (HDAC3-WT), expression of ABD-deletion mutant of HDAC3 failed to inhibit AKT acetylation (Fig 3H). It also lost the ability to increase polyubiquitination and phosphorylation of AKT (Fig 3I and J). Furthermore, by examining more than 100 AKT-interacting proteins, we noticed that protein sequences in the AKT-binding region are reported only in five of them (APPL1, YB-1, BRCA1, MEN1, and DAB2), but we found no consensus AKT-binding sequence between these five proteins and HDAC3, suggesting that the ABD in HDAC3 is unique. Together, our data demonstrate that HDAC3 interacts with AKT via ABD, which is indispensable for HDAC3-enhanced AKT phosphorylation.

### HDAC3 associates with AKT in plasma membrane and promotes lysine-63 polyubiquitination of AKT

Plasma membrane association is critical for AKT phosphorylation and activation (Gao et al, 2011). A significant percentage of HDAC3 proteins are localized in the plasma membrane (Wen et al, 2003;

Longworth & Laimins, 2006). In concordance with our findings in prostate cancer cells, a recent chemoresistance study showed that HDAC3 regulates AKT phosphorylation in acute myeloid leukemia (Long et al, 2017). Immunofluorescent cytochemistry (IFC) staining in the same study indicated that HDAC3 and AKT interact in the nucleus, suggesting that HDAC3 regulates AKT phosphorylation through its nuclear function. In contrast, our confocal microscopy analysis showed that both proteins were detectable in the plasma membrane, cytoplasm, and nucleus of both LNCaP and C4-2 prostate cancer cells (Figs 4A and EV2A). Moreover, protein fractionation assays showed that endogenous HDAC3 and AKT proteins interacted with each other in both cytoplasmic and nuclear compartments (Fig 4B). Most importantly, we demonstrated that increased expression of HDAC3 markedly enhanced AKT association with plasma membrane, but not in the nucleus (Fig EV2B). Furthermore, we generated a C-terminal truncated mutant of HDAC3 [HDAC3 (1–313)] (Fig EV2C), which has been shown to be localized mainly in the plasma membrane and cytoplasm (Yang et al, 2002). We confirmed it is also the case in C4-2 cells (Fig EV2D and E). Most importantly, ectopic expression of the HDAC3 cytoplasmic mutant enhanced IGF-1-induced AKT phosphorylation to the extent similar to the wild-type counterpart (Fig 4C). These data suggest that the cytoplasmic function of HDAC3 is important for its regulation of AKT phosphorylation.

To determine whether HDAC3 regulation of AKT acetylation requires its deacetylase activity, we constructed HDAC3-H134Q, a deacetylase inactivation mutant of HDAC3 (Wen et al, 2003). We demonstrated that apart from HDAC3-WT, expression of H134Q mutant failed to diminish AKT acetylation (Figs 3H and 4D). The same was true for AKT polyubiquitination and phosphorylation (Figs 3I and J, and 4E). It is worth noting that the deacetylation enzymatic activity of HDAC3 is important for its role in regulating gene expression. However, whether the enzymatic dead mutant H134Q and the AKT-binding deficient mutant AKTΔABD have common or distinct effects on gene expression is unclear at present and warrants further investigation.

Lysine residues 14 and 20 (K14 and K20) were identified as two major acetylation sites on AKT (Pillai et al, 2014). Authors of a recent study in leukemia cells concluded that HDAC3 activates AKT by deacetylating AKT at K20 (Long et al, 2017). However, data presented in the study clearly showed that expression of K20R mutant of HDAC3 markedly increased AKT phosphorylation at both T308 and S473 sites (Long et al, 2017), arguing the existence of additional lysine residue(s) in facilitating AKT ubiquitination and phosphorylation. Indeed, our mutagenesis studies demonstrated that only dual acetylation-mimicking mutant of K14 and K20 (K14Q and K20Q), but not individual mutant alone, abolished HDAC3-augmented polyubiquitination and phosphorylation of AKT in C4-2 cells (Fig 4F and G). Since lysine 63 (K63)-chain polyubiquitination is important for membrane localization and phosphorylation of AKT (Pickart, 2001), we examined the effect of HDAC3 on AKT K63-chain polyubiquitination. We found that expression of HDAC3 increased AKT polyubiquitination and this effect was abolished by K63R-mutated Ub, but not K48R or WT Ub (Fig 4H). These data suggest that deacetylation at both K14 and K20 residues is important for HDAC3 regulation of AKT phosphorylation and this effect of HDAC3 is mediated by K63-linked polyubiquitination of AKT.

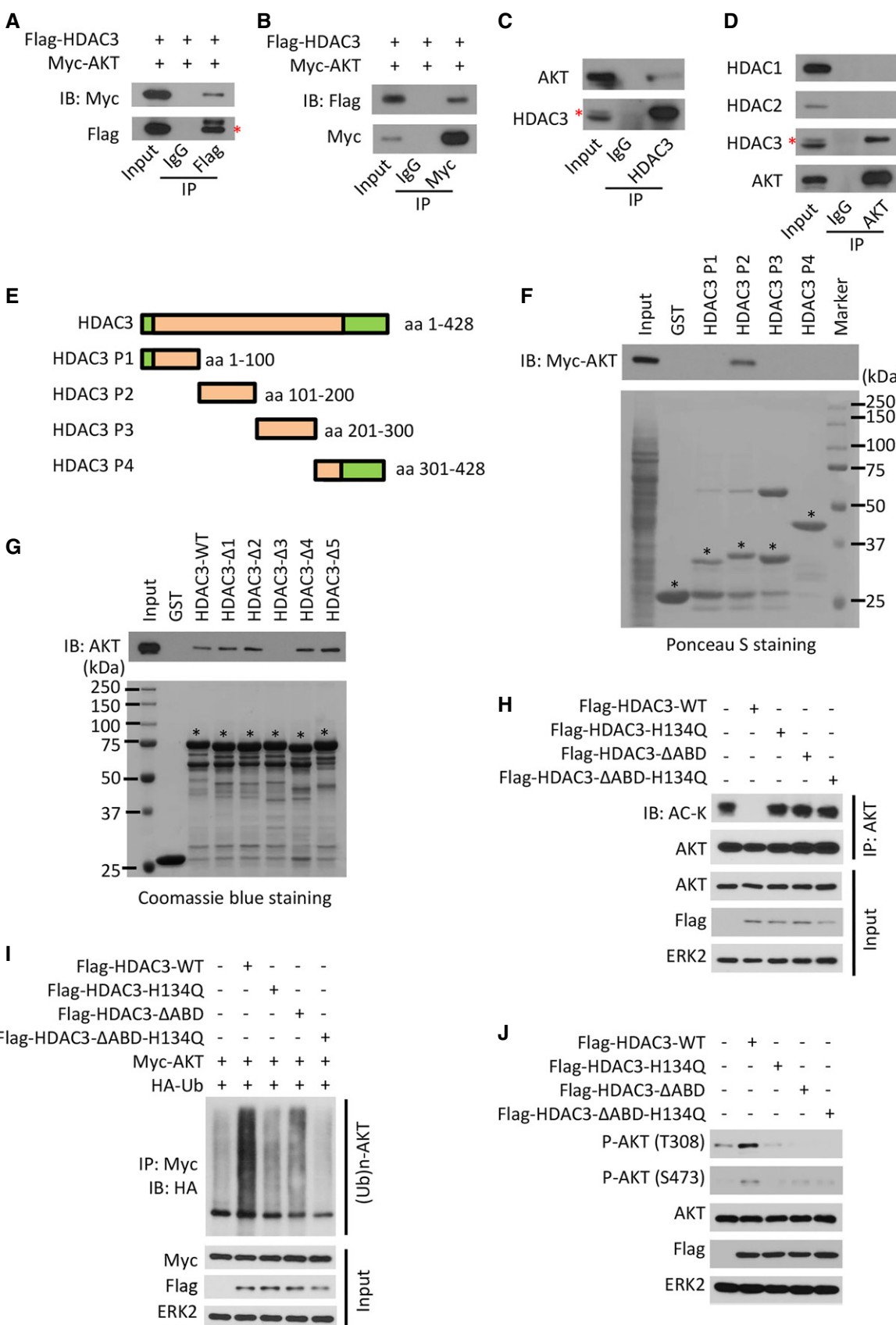

Figure 3.

**Figure 3.  A region (aa 141–160) in HDAC3 is indispensable for its interaction with AKT.**

A, B    293T cells were transfected with indicated plasmids for 24 h followed by IP and Western blots with the indicated antibodies. The asterisk (*) indicates the specific
        Flag-HDAC3 protein band.
C, D    C4-2 cell lysate was prepared for IP and Western blots with the indicated antibodies. The asterisk (*) indicates the specific Flag-HDAC3 protein band.
E       An illustration depicts four fragments of HDAC3 protein for generation of GST recombinant proteins.
F, G    C4-2 cells were transfected with Myc-tagged AKT for 24 h, and cell lysate was prepared for GST pull-down assay. Input and GST or GST-HDAC3 recombinant
        proteins used for pull-down assay in (F) were examined by Ponceau S staining and pull-down proteins were evaluated by Western blot with anti-Myc antibody.
        GST or GST-HDAC3 recombinant proteins used for pull-down assay in (G) were examined by Coomassie blue staining. GST or GST-HDAC3 recombinant proteins
        with expected molecular mass are indicated by asterisks.
H–J     C4-2 cells were transfected with indicated plasmids for 24 h followed by IP and Western blots with the indicated antibodies.

Source data are available online for this figure.

## APPL1 facilitates HDAC3-AKT interaction and AKT activation

Overexpression of the scaffold protein APPL1 stimulates insulin-mediated AKT phosphorylation (Mao *et al*, 2006; Deepa & Dong, 2009), highlighting the role of APPL1 expression in insulin signal transduction; however, its precise underlying mechanism is unclear. We therefore sought to investigate whether APPL1 plays any role in HDAC3 regulation of AKT. Co-IP assay demonstrated that endogenous HDAC3 and APPL1 interacted with one another in C4-2 cells (Fig 5A). GST pull-down assay further showed that the P1 fragment (aa 1–100) of HDAC3 was specifically bound by APPL1 (Figs 3E and 5B); this is different from the ABD motif located in the P2 fragment (Fig 3E and F). To define the specific fragment in APPL1 required for HDAC3 binding, we constructed five GST-APPL1 recombinant proteins corresponding to four well-studied functional domains of APPL1. GST pull-down assays indicated that the PH domain is the only region specifically interacting with HDAC3 (Fig 5C and D).

Although knockdown of APPL1 by siRNAs did not affect the expression level of HDAC3 and AKT proteins (Fig 5E), it largely diminished AKT binding with HDAC3 and AKT phosphorylation at both T308 and S473 (Fig 5E and F). Given that HDAC3 is a key component of the NCOR/SMRT complex, we examined the effect of APPL1 knockdown on this complex. We demonstrated that APPL1 knockdown largely increased HDAC3 interaction with both NCOR and SMRT (Fig EV2F), suggesting that APPL1 binding can prevent NCOR/SMRT from binding to HDAC3. Moreover, IGF-1 stimulation increased HDAC3 binding with AKT and APPL1 and enhanced APPL1 binding with AKT and HDAC3 (Fig 5G and H), suggesting that these three proteins cooperate in IGF-1 transduced signaling. Importantly, depletion of HDAC3 by siRNAs abolished both basal level and IGF-1-enhanced interaction between APPL1 and AKT (Fig 5H), highlighting the importance of HDAC3 in mediating the interaction between AKT and APPL1. Furthermore, IGF-1 stimulation increased IGFR engagement with APPL1, HDAC3, and AKT (Fig 5I). Most strikingly, silencing of APPL1 disrupted IGFR association with AKT and HDAC3, whereas depletion of HDAC3 only disrupted IGFR interaction with AKT, but not APPL1 (Fig 5I and J). Based upon these findings, we envision a model in which, in the absence of activation of IGFR by IGF-1, APPL1 and HDAC3 drift in the cytoplasm and AKT is subject to be acetylated and thereby immune to E3 ligase-mediated ubiquitination (Fig 5K). Upon the stimulus of IGF-1, IGFR recruits APPL1 which in turn scaffolds the interaction between HDAC3 and AKT and facilitates the deacetylation of AKT by HDAC3, making AKT poised for polyubiquitination, phosphorylation, and activation (Fig 5K).

## *Hdac3* deletion attenuates *Pten* deletion-mediated prostate tumorigenesis

Approximately 70% of prostate cancers lose one copy of *PTEN* gene by the time of diagnosis (Chen *et al*, 2005b). PTEN loss leads to AKT hyperactivation and prostate tumorigenesis and progression (Majumder & Sellers, 2005; Shukla *et al*, 2007). As expected, AKT phosphorylation was elevated by PTEN knockdown in PTEN-positive 22Rv1 prostate cancer cells, but PTEN loss-enhanced AKT phosphorylation was mitigated by HDAC3 co-knockdown (Fig 6A). Knockdown of HDAC3 in PTEN-negative C4-2 cells decreased cell growth in both 2-dimension (2D) and 3D cultures (Fig EV3A–E). Confocal microscopy analysis indicated that the growth inhibitory effect was correlated with decreased AKT phosphorylation (Fig EV3F and G). These data indicate an important role of HDAC3 in PTEN loss-mediated AKT activation.

To extend our observation from *in vitro* to *in vivo* studies, prostate-specific *Pten* homozygous deletion mouse model was employed. This mouse model recapitulates prostate tumorigenesis and progression and is considered as a reliable and valuable model to study prostate cancer (Lesche *et al*, 2002; Wang *et al*, 2003). To investigate the effect of *Hdac3* loss on Akt phosphorylation and associated prostate tumorigenesis, we crossbred probasin-Cre transgenic mice (*Pb-Cre4*; Wu *et al*, 2001) with *Hdac3* conditional (*Hdac3*$^{Loxp/Loxp}$, *Hdac3*$^{L/L}$; Bhaskara *et al*, 2008) and *Pten* conditional (*Pten*$^{Loxp/Loxp}$, *Pten*$^{L/L}$) mice (Wang *et al*, 2003). A cohort of mice with four different genotypes were generated: prostate-specific *Pten* knockout alone (*Pb-Cre4*; *Pten*$^{L/L}$, hereafter termed as *Pten*$^{pc-/-}$), *Pten/Hdac3* double knockout (*Pb-cre4*; *Pten*$^{L/L}$;*Hdac3*$^{L/L}$, termed as *Pten*$^{pc-/-}$; *Hdac3*$^{pc-/-}$), *Hdac3* knockout alone (*Pb-Cre4*; *Hdac3*$^{L/L}$, termed as *Hdac3*$^{pc-/-}$), and Cre-negative *Pten*$^{L/L}$; *Hdac3*$^{L/L}$ control mice (termed as "wild-type").

As demonstrated by IHC, Hdac3 protein was readily detected in the prostates of "wild-type" and *Pten*$^{pc-/-}$ mice, but little to no Hdac3 protein was detected in the prostates of *Hdac3*$^{pc-/-}$ and *Pten*$^{pc-/-}$;*Hdac3*$^{pc-/-}$ counterparts (Fig 6B-i). As expected, Pten protein was barely detected in *Pten*$^{pc-/-}$ and *Pten*$^{pc-/-}$;*Hdac3*$^{pc-/-}$ prostates, while it was readily expressed in "wild-type" and *Hdac3*$^{pc-/-}$ prostates (Fig 6B-ii). Consistent with the results of *Pten* deletion in the cell culture model (Fig 6A), Akt phosphorylation level was robustly increased in the prostates of *Pten*$^{pc-/-}$ mice compared with Pten-positive controls, while *Hdac3* loss significantly diminished AKT phosphorylation in prostate tumors with *Pten*$^{pc-/-}$ background (Fig 6B-iii). In accordance with IHC results shown in Fig 6B-iii, Western blot analysis also revealed that Akt

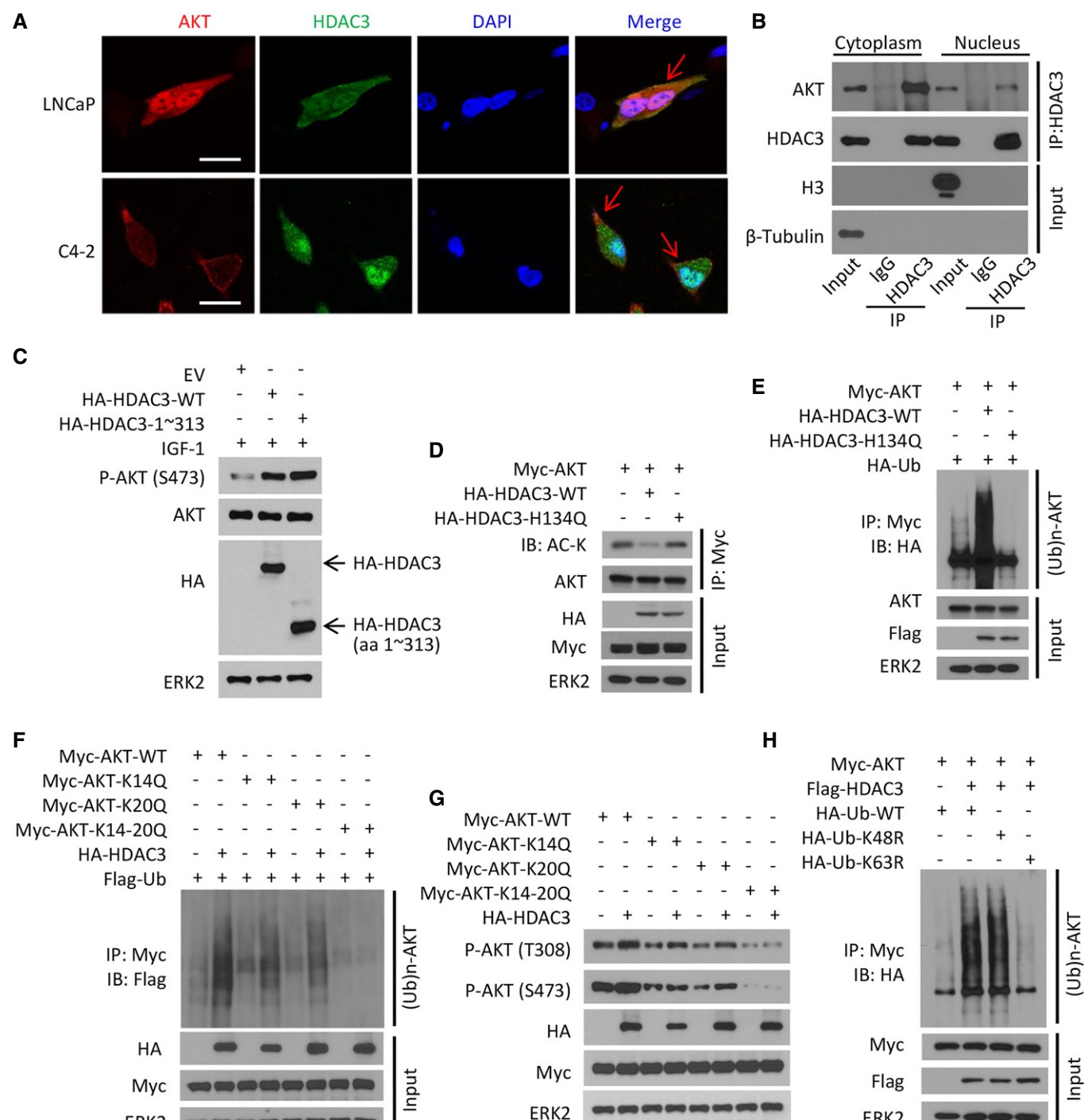

**Figure 4. HDAC3 associates with AKT in plasma membrane and induces lysine 63-linked ubiquitination of AKT.**

A    LNCaP and C4-2 cells grown in logarithmic phase were fixed and subjected to IFC. Cell nuclei were counterstained by DAPI. Arrows point to the co-localization of AKT and HDAC3 proteins in the plasma membrane. Approximately 80% of LNCaP cells and 85% of C4-2 cells showed the co-localization of these two proteins on plasma membrane. Scale bars, 20 μm.

B    C4-2 cells were treated with IGF-1 for 30 min and then harvested for cellular fractionation followed by IP and Western blots with the indicated antibodies.

C    C4-2 cells were transfected with empty vector (EV) or HA-tagged wild-type or mutant HDAC3 for 24 h followed by treatment of 10 ng/ml IGF-1 for 30 min. Cells were harvested for Western blots with the indicated antibodies.

D–H  C4-2 cells were transfected with the indicated plasmids for 24 h and harvested for IP and Western blots with the indicated antibodies.

Source data are available online for this figure.

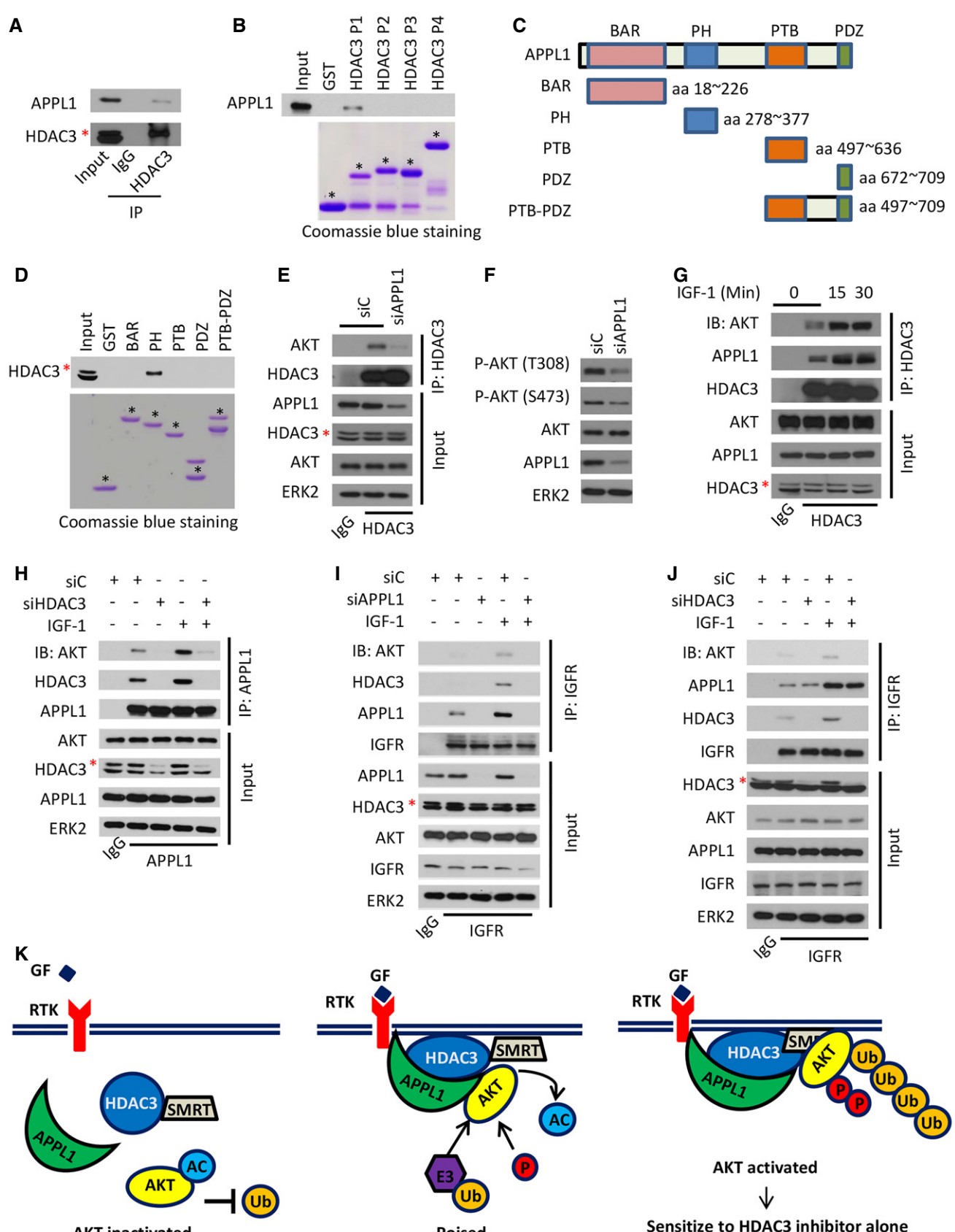

Figure 5.

**Figure 5.  The scaffold protein APPL1 facilitates HDAC3 regulation of AKT.**

A    C4-2 cell lysate was prepared for IP and Western blots with the indicated antibodies. The asterisk (*) indicates the specific HDAC3 protein band.

B    C4-2 cell lysate was prepared for GST pull-down assay using GST or GST-HDAC3 recombinant proteins (stained with Coomassie blue, low panel) followed by Western blot with anti-APPL1 antibody (upper panel). GST or GST-HDAC3 recombinant proteins with expected molecular mass are indicated by asterisks.

C    An illustration depicts four functional domains (BAR, PH, PTB, and PDZ) of APPL1 used for construction of GST-APPL1 recombinant proteins.

D    C4-2 cell lysate was prepared for GST pull-down assay using GST or GST-APPL1 recombinant proteins (stained with Coomassie blue, low panel) followed by Western blot with anti-HDAC3 antibody (upper panel). The red asterisk (*) indicates the specific HDAC3 protein band, while the black ones indicate the specific domains of APPL1.

E, F    C4-2 cells were transfected with a pool of control or APPL1-specific siRNAs for 48 h followed by IP and Western blots with the indicated antibodies. The asterisk (*) indicates the specific HDAC3 protein band.

G    C4-2 cells were treated with 10 ng/ml of IGF-1 for different periods of time followed by IP and Western blots with the indicated antibodies. The asterisk (*) indicates the specific HDAC3 protein band.

H–J    C4-2 cells were transfected with indicated siRNAs and treated with 10 ng/ml of IGF-1 for 30 min and followed by IP and Western blots with the indicated antibodies. The asterisks (*) indicate the specific HDAC3 protein bands.

K    A hypothetical model depicting roles of HDAC3 and APPL1 in growth factor (GF)-induced AKT activation. In the absence of the interaction of GF with a receptor tyrosine kinase (RTK), HDAC3 and APPL1 drift around in the cytosol. As a result, AKT becomes highly acetylated and resistant to be polyubiquitinated. Upon GF stimulation, RTK recruits APPL1, which in turn functions as a scaffold facilitating HDAC3-mediated deacetylation of AKT, thereby making AKT poised for further activation by polyubiquitination. Activation of this deacetylase-dependent function of HDAC3 may also require the binding by the deacetylase activating domain of SMRT.

Source data are available online for this figure.

phosphorylation was largely decreased in *Hdac3* and *Pten* double knockout tumor tissues compared to *Pten* single knockout tumors (Fig 6C). Meanwhile, Akt acetylation was elevated due to *Hdac3* deletion (Fig 6C), further supporting the conclusion that *Hdac3* loss undermined Akt phosphorylation by increasing its acetylation.

By following up on the survival of a cohort of 83 mice for over 12 months, we found that mice in both "wild-type" and $Hdac3^{pc-/-}$ groups all survived (Fig 6D). Notably, three mice died in the $Pten^{pc-/-};Hdac3^{pc-/-}$ group, whereas seven mice died in the $Pten^{pc-/-}$ group (Fig 6D), suggesting that conditional knockout of *Hdac3* significantly prolonged the overall survival of mice with a $Pten^{pc-/-}$ background. As expected, both "wild-type" and $Hdac3^{pc-/-}$ did not display visible tumors in the prostate, while the prostate lobes of $Pten^{pc-/-};Hdac3^{pc-/-}$ mice at 4 months old were obviously smaller than those of age-matched $Pten^{pc-/-}$ mice, indicating *Hdac3* loss delays the growth of tumors in $Pten^{pc-/-}$ mice (Fig 6E and F).

Prostatic intraepithelial neoplasia (PIN) is a commonly used indicator of prostate tumorigenesis in mouse models (Park *et al*, 2002). We analyzed the frequency of PIN in all the lobes and found that the majority of acini were high-grade PIN (HGPIN)/cancer in the prostates of $Pten^{pc-/-}$ mice but more low-grade PIN (LGPIN) in the $Pten^{pc-/-};Hdac3^{pc-/-}$ group (Fig 6F and G). Since Akt phosphorylation is associated with cell proliferation and survival (Lawlor & Alessi, 2001), we used proliferation marker Ki67 to evaluate cell proliferation in prostate tissues. By performing Ki-67 IHC and quantifying the percentage of Ki67-positive cells, we demonstrated that *Hdac3* loss reduced the percentage of proliferative cells in Pten-deficient prostates (Fig 6H and I). In contrast, cleaved caspase-3 IHC analysis demonstrated that Hdac3 loss had no overt effect on apoptosis (Fig EV4A and B). These data suggest that the inhibitory effect of Hdac3 loss on *Pten* deletion-induced tumorigenesis in the prostate is primarily mediated by decreased cell proliferation, but less likely via its effect on apoptosis. These data indicate that HDAC3 plays a pivotal role in PTEN loss-induced AKT activation and prostate tumorigenesis and progression.

Cell culture studies have shown previously that pharmacological inhibition or knockdown of HDAC3 blocks AR-mediated transcriptional activation of its target genes, such as *TMPRSS2* and *NKX3.1*, although the underlying mechanisms remain poorly understood

**Figure 6.  *Hdac3* deletion decreases AKT phosphorylation and tumor growth in *Pten* knockout prostate cancer.**

A    22Rv1 cells were transfected with a pool of control and gene-specific siRNAs for 48 h followed by Western blots with the indicated antibodies. The asterisk (*) indicates the specific HDAC3 protein band.

B    IHC for Hdac3 (i), Pten (ii) and phosphorylated Akt (p-Akt-S473) (iii) in prostate tissues of "wild-type", $Hdac3^{pc-/-}$, $Pten^{pc-/-}$, and $Pten^{pc-/-};Hdac3^{pc-/-}$ mice at age of 4 months. The inset is a high-magnification image of the framed area in each panel. Scale bar: 50 μm; scale bar in the inset: 10 μm.

C    Protein samples were prepared from prostate tissues of $Pten^{pc-/-}$ and $Pten^{pc-/-};Hdac3^{pc-/-}$ mice at age of 4 months followed by IP and Western blots with the indicated antibodies.

D    Kaplan–Meier survival analysis of "wild-type" (n = 25), $Hdac3^{pc-/-}$ (n = 18), $Pten^{pc-/-}$ (n = 15), $Pten^{pc-/-};Hdac3^{pc-/-}$ (n = 25) mice. The "n" indicates the number of mice. *P = 0.012 comparing the overall survival of $Pten^{pc-/-};Hdac3^{pc-/-}$ mice (n = 25) with "$Pten^{pc-/-}$" mice (n = 15) by Gehan–Breslow–Wilcoxon test.

E    Representatives of genitourinary tracts of $Pten^{pc-/-}$ and $Pten^{pc-/-};Hdac3^{pc-/-}$ mice at age of 4 months.

F    Representatives of H&E staining for ventral prostate (VP) of 4-month-old mice with indicated genotypes. Scale bar: 50 μm.

G    Quantification of non-malignant, low-grade PIN (LGPIN), and high-grade PIN (HGPIN)/cancer acini in the lobes of AP, VP, and DLP of the mice with the indicated genotypes (n = 6 mice for each group).

H, I    Representatives of Ki67 staining in prostate tissues from $Pten^{pc-/-}$ and $Pten^{pc-/-};Hdac3^{pc-/-}$ mice at age of 4 months are shown in (H) with the quantitative data in (I). Scale bar: 100 μm; scale bar in the inset: 10 μm. Data are shown as means ± SEM (n = 6 mice for each group); ***P = 1.64e-04 was performed by the unpaired two-tailed Student's t-test.

Source data are available online for this figure.

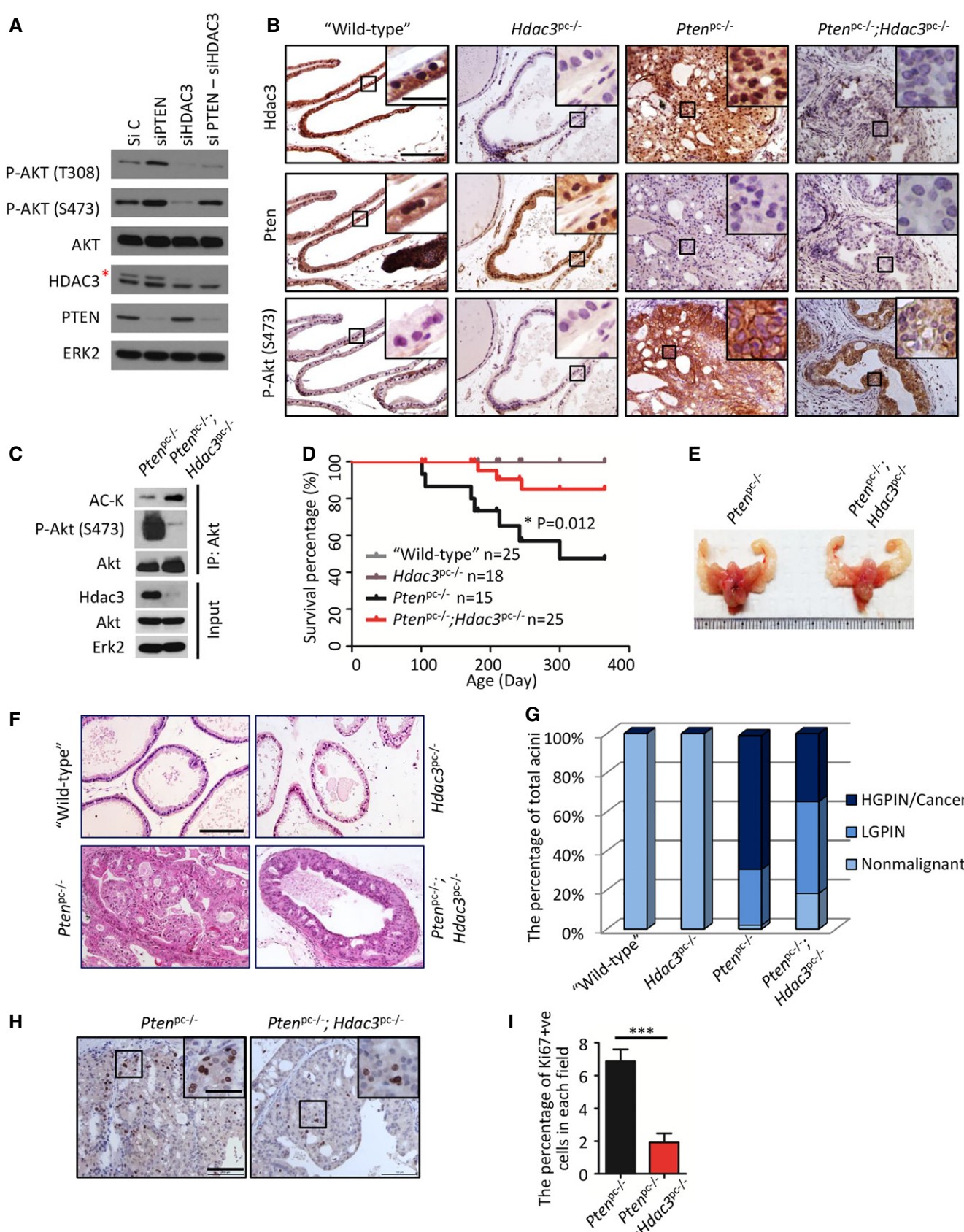

**Figure 6.**

(Welsbie *et al*, 2009). In the *Hdac3* knockout mouse prostate, we demonstrated that homozygous deletion of *Hdac3* not only decreased Ar protein level, but also decreased expression of its target genes including *Probasin*, *Tmprss2*, and *Nkx3.1* in both wild-type and *Pten*-deficient backgrounds (Fig EV4C and D). We further showed that ectopic expression of wild-type HDAC3 increased expression of AR target genes *NXK3.1* and *PSA* in LNCaP cells, and similar results were obtained in cells transfected with HDAC3 (1–313), a C-terminal truncated mutant of HDAC3 that is localized mainly in the plasma membrane and cytoplasm (Figs EV2D and E, and EV4E–G). These data indicate that HDAC3 regulates AR transcriptional activity in prostate cancer cells in culture and in mice and this effect is likely mediated by its function in the cytoplasm. It is worth noting that AR is a transcription factor which is active in the nucleus, and therefore, it is possible that the cytosolic HDAC3 may modulate AR activity through indirect regulation of AR-regulatory signaling pathways.

## HDAC3 inhibitor RGFP966 inhibits growth of PTEN-deficient prostate cancer cell lines, organoids, and xenografts

Given that selective HDAC3 inhibitor RGFP966 exhibited a pronounced anti-cancer effect in hematological malignancies (Matthews *et al*, 2015), we sought to determine whether RGFP966 inhibits cell growth of solid tumors such as prostate cancer. A Western blot-based survey was conducted to identify suitable cell lines for study. C4-2 and 22Rv1 were selected for further analysis because (i) both cell lines are AR-positive, (ii) both expressed similar levels of HDAC3, and (iii) one is PTEN-negative and the other is PTEN-positive (Fig 7A). Both cell lines were treated with RGFP966 or AKT inhibitor GDC0068, an ATP-competitive pan-AKT inhibitor that has been used for treatment of solid tumors in a phase I trial

(Saura *et al*, 2017). Clonogenic survival assay demonstrated that PTEN-null C4-2 cells were more sensitive to GDC0068 compared with PTEN-positive 22Rv1 cells (Fig 7B). This result is consistent with a previous report that cell lines with a high level of AKT phosphorylation are more sensitive to GDC0068 (Lin *et al*, 2013), a notion of "oncogene addiction". On the contrary, we observed that 22Rv1 (IC50: 0.720 μM) was more sensitive to RGFP966 compared with C4-2 (IC50: 1.19 μM) (Fig 7C), indicating that PTEN status is a potential determinant of the efficacy of HDAC3 inhibitor in prostate cancer treatment. Interestingly, the steep curves of the clonogenic survival data clearly showed that both cell lines were more sensitive to RGFP966 than GDC0068 (Fig 7B and C), suggesting that prostate cancer cells are vulnerable to HDAC3 inhibition. Based upon these observations, we predicted that a higher concentration of RGFP966 alone should be more efficacious in inhibition of C4-2 cell growth than GDC0068. To test this hypothesis, we treated C4-2 cells with GDC0068 or RGFP966 at the concentration of 1× or 2× IC50. Clonogenic survival assay showed that the higher concentration of RGFP966 markedly inhibited cell growth with formation of fewer and smaller colonies, while the higher dose of GDC0068 only slightly decreased the size and number of colonies (Fig 7D and E).

S6K and 4E-BP1 phosphorylation are two well-defined downstream events of AKT activation (Chung *et al*, 1994). We demonstrated that S6K phosphorylation was only partially diminished by the higher dose of GDC0068 in C4-2 cells (Fig 7F). Most strikingly, the treatment with a higher concentration of RGFP966 totally inhibited S6K phosphorylation (Fig 7F). A similar trend was observed for 4E-BP1 phosphorylation even though the inhibitory effect of GDC0068 on 4E-BP1 phosphorylation was more pronounced than that on S6K phosphorylation (Fig 7F). Notably, similar to previous reports related to AKT-mTOR inhibitors (Carver *et al*, 2011; Mulholland *et al*, 2011), we also found that GDC0068 treatment increased

---

**Figure 7.  The HDAC3 inhibitor suppresses PTEN-deficient prostate cancer growth.**

A    Cell lysate was prepared from indicated prostate cancer cell lines for Western blot analysis. Arrows show the full length and variants of AR.

B, C    C4-2 and 22Rv1 cells were treated with GDC0068 (B) and RGFP966 (C). IC50 is shown as a dotted line in the middle of the graph; for C4-2 cells, IC50 of GDC0068 = 1.85 μM; IC50 of RGFP966 = 1.19 μM; for 22Rv1 cells, IC50 of GDC0068 = 2.67 μM; IC50 of RGFP966 = 0.72 μM. The survival curve was generated from three independent experiments and each experiment was in triplicate. The error bars indicate the smallest and largest value among three independent experiments, which represented by lower whisker and upper whisker, respectively.

D, E    C4-2 cells were treated with low (L, 1× IC50) or high (H, 2× IC50) concentrations of GDC0068 or RGFP966 for 4 days. The number of colonies with more than 50 cells was counted. Representatives of colonies are shown in (D) with quantification data shown in (E). Data represent means ± SEM; GDC (L) versus GDC (H): *$P$ = 0.037; GDC0068 (H) versus RGFP (H): ***$P$ = 1.47e-05, RGFP (L) versus RGFP (H): **$P$ = 3.55e-05 were performed by the unpaired two-tailed Student's *t*-test.

F    C4-2 cells were treated with GDC0068 or RGFP966 for 24 h followed by Western blots.

G, H    Representative images of 3D cultures of C4-2 cells at day 5 post-treatment of GDC0068 (H), RGF966 (L), or RGF966 (H) as shown in (G) with quantification data shown in a box plot (H). Each box in the graph indicates the interquartile range (IQR). The horizontal line represents the median value. Box lower limit is the first quantile (Q1) while the upper limit is the third quantile (Q3). The lower whisker is max(min(x)), Q1 − 1.5*IQR while the upper whisker is min(max(x)), Q3 + 1.5*IQR. DMSO ($n$ = 144) versus GDC (H) ($n$ = 180): ***$P$ = 2.23e-32, DMSO versus RGFP (L) ($n$ = 233): ***$P$ = 1.04e-36 and DMSO versus RGFP (H) ($n$ = 83): ***$P$ = 3.12e-35 were performed by Wilcoxon rank sum test with continuity correction. Scale bar, 100 μm.

I, J    Mice with C4-2 xenograft tumors were treated with vehicle (DMSO), GDC0068 (H) (50 mg/kg), RGFP966 (L) (25 mg/kg), or RGFP966 (H) (50 mg/kg) 5 days a week for three consecutive weeks (I). Images of tumors isolated at day 21 are shown in (J). Data are shown as means ± SEM. DMSO ($n$ = 7) versus GDC0068 (H) ($n$ = 7): ***$P$ = 4.18e-10, DMSO versus RGFP966 (L) ($n$ = 7): ***$P$ = 1.40e-09, GDC0068 (H) versus RGFP966 (H) ($n$ = 7): *$P$ = 0.0182, RGFP966 (L) versus RGFP966 (H): *$P$ = 0.0104 comparing the tumor volume at day 21 post-treatment by the unpaired two-tailed Student's *t*-test.

K–M    Representatives of organoids at day 5 post-treatment of GDC0068 (H), RGF966 (L), or RGF966 (H) are shown in (K) with quantification data of OD value at 490 nm in (L). The OD value was measured and quantified from three biological replicates. Data represent means ± SEM; DMSO versus GDC0068 (H): ***$P$ = 2.18e-05; DMSO versus RGFP966 (L): ***$P$ = 1.43e-04, DMSO versus RGFP966 (H): ***$P$ = 1.40e-08 were performed using the unpaired two-tailed Student's *t*-test. Based on the observed growth rate of untreated PTEN-deleted organoids, greater than 50% of organoids reach 30 μm in diameter at day 5. "30 μm" was set as the cutoff value. The number of organoids with the diameter > 30 μm from at least five fields (each field contains at least 7 organoids) were counted and analyzed from three biological replicates (M). Data are shown as means ± SEM; DMSO versus GDC0068 (H): ***$P$ = 8.02e-05, DMSO versus RGFP966 (L): ***$P$ = 1.39e-04, DMSO versus RGFP966 (H): ***$P$ = 2.96e-06 were performed by the unpaired two-tailed Student's *t*-test. Scale bar, 100 μm.

Source data are available online for this figure.

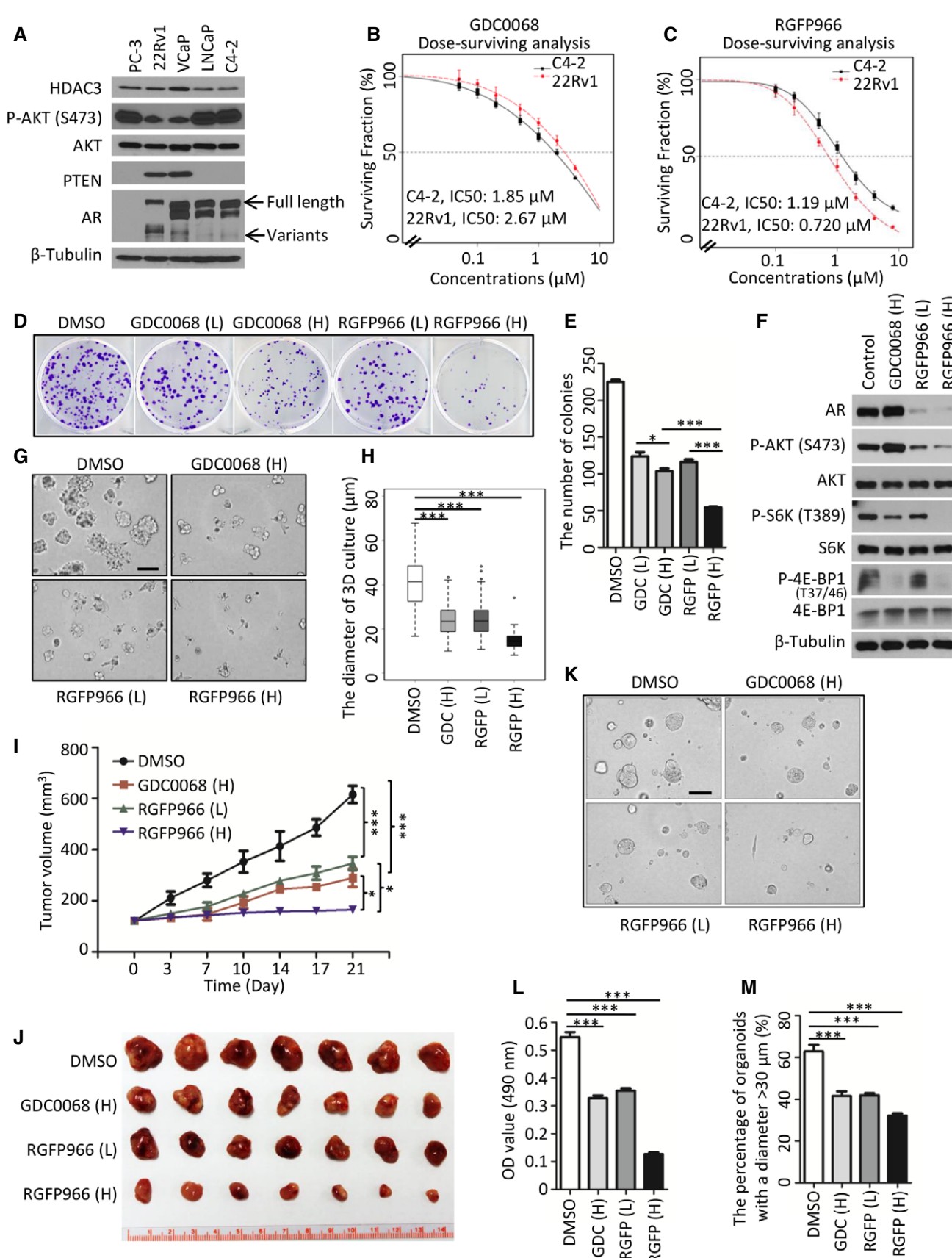

**Figure 7.**

AR protein level (Fig 7F), providing a plausible explanation as to why the cells were relatively resistant to GDC0068 treatment. In contrast, we found that, in addition to inhibiting AKT signaling, RGFP966 inhibited expression of AR and its downstream target genes including *NKX3.1*, *PSA*, and *TMPRSS2* in C4-2 cells (Figs 7F, and EV5A and B). This is consistent with a previous finding showing that depletion of HDAC3 by shRNA diminished AR mRNA level (Welsbie *et al*, 2009). To further clarify the contribution of AKT phosphorylation inhibition or AR reduction to the overall efficacy of RGFP966, AR and constitutively active (myristoylated) AKT (CA-AKT) were exogenously expressed in C4-2 cells and drug sensitivity was examined in these cells. Clonogenic survival assays showed that exogenous expression of AR or CA-AKT alone only partially blocked RGFP966-induced inhibition of C4-2 cell growth (Fig EV5C and D), suggesting that both AKT inhibition and AR reduction are important for RGFP966 inhibition of cell growth. The 3D Matrigel culture further confirmed the anti-cancer effect of the higher concentration of RGFP966 on C4-2 cell growth (Fig 7G and H). Studies with C4-2 xenograft model further showed that a higher concentration of RGFP966 was efficient in inhibiting tumor growth in mice (Fig 7I and J).

To further extend our findings from cell culture systems to more clinically relevant models, a PTEN-deficient patient-derived organoid line (Gao *et al*, 2014) was employed. In this model, we also observed that the higher concentration of RGFP966 not only significantly impaired the overall proliferation of organoids cultured in 3D (Fig 7K and L), but also significantly decreased the size of organoids (Fig 7M). Together, these data indicate that the HDAC3 inhibitor blocks PTEN-mutated prostate cancer cell growth by targeting both AKT and AR signaling.

### Inhibition of HDAC3 suppresses *SPOP*-mutated prostate cancer growth

Approximately 10% of prostate cancer patients harbor *SPOP* mutations (Barbieri *et al*, 2012). In this subtype of prostate cancers, both AKT and AR signaling are aberrantly activated (An *et al*, 2014; Geng *et al*, 2014; Blattner *et al*, 2017; Zhang *et al*, 2017). Currently, strategies for effectively treating this unique subclass of patients remain to be explored. Given that inhibition of HDAC3 can target both AKT and AR signaling pathways, we hypothesized that RGFP966 is an effective agent to suppress the growth of *SPOP*-mutated tumors. To mimic the scenario in patient samples where SPOP mutations detected so far are heterozygous and almost mutually exclusive with PTEN mutations or deletions, we introduced SPOP-mutant F133V, a hotspot mutation (Barbieri *et al*, 2012), into a PTEN-positive 22Rv1 cell line in which AKT phosphorylation level is relatively low (Fig 7A). Similar to the findings reported recently (An *et al*, 2014; Blattner *et al*, 2017; Zhang *et al*, 2017), we observed that expression of SPOP F133V elevated the level of phosphorylated AKT, S6K, and 4E-BP1 as well as that of the full-length AR, but had almost no effect on AR splice variants (Fig 8A). However, following RGFP966 treatment, phosphorylation of AKT and its downstream targets S6K and 4E-BP1 were significantly decreased in SPOP-F133V-expressing 22Rv1 cells (Fig 8A). Notably, the expression of both full-length AR and variants was almost completely inhibited by RGFP966 in a dose-dependent manner (Figs 8A and EV5B).

To investigate whether HDAC3 inhibitor alone enables to suppress *SPOP*-mutant cell growth, both empty vector (EV)- and SPOP-F133V-expressing 22Rv1 cells in 3D cultures were treated with RGFP966. Similar to the results in C4-2 cells (Fig 7G), RGFP966 alone was efficient to inhibit growth of SPOP-F133V-expressing 22Rv1 cells (Fig 8B and C). The treatment also effectively inhibited growth of patient-derived *SPOP*-mutated organoids (Fig 8D and E). The inhibitory effect of RGFP966 was further confirmed by the significant decrease in the size of organoids (Fig 8F). The IFC results demonstrated that RGFP966 effectively inhibited both AR and AKT signaling in these organoids (Fig 8G and H). Thus, our data suggest that inhibition of HDAC3 effectively suppresses the growth of SPOP-mutated prostate cancer cells by shutting down both AKT and AR signaling pathways.

## Discussion

AKT is abnormally activated in up to 70% of advanced prostate cancers due to frequent deletion of *PTEN* or active mutations in PI3K/AKT pathway genes (Taylor *et al*, 2010). Deregulation of PI3K/PTEN pathway results in aberrant elevation of phosphatidylinositol (3,4,5)-trisphosphate (PIP3) and subsequent phosphorylation of AKT mediated by PDK1 and mTORC2. Besides phosphorylation, polyubiquitination of lysine residues on AKT and membrane transportation is also required for its activation (Yang *et al*, 2009). In contrast, AKT phosphorylation can be suppressed by pan inhibitors of class I/II HDACs (Chen *et al*, 2005a). A possible explanation of this observation is that pan inhibitor treatment results in increased histone acetylation which promotes the expression of genes encoding such proteins as PHLPP1, a protein phosphatase that decreases AKT phosphorylation in chondrocytes (Bradley *et al*, 2013). However, we provided evidence that inhibition of *de novo* protein synthesis by CHX failed to have overt effect on class I/II HDACI-induced suppression of AKT phosphorylation in prostate cancer cells and the drug effect appears to be an immediate event (30 min post-treatment). These findings suggest that inhibition of AKT phosphorylation by class I/II HDACIs is not primarily mediated by the genomic effect of these HDACIs in the nucleus of prostate cancer cells. Class I/II HDACI treatment studies also raise a very important question as to which HDAC protein(s) mediate the effect on AKT phosphorylation. Through unbiased shRNA screening, we demonstrated for the first time that HDAC3 is the only member of the class I/II HDAC subfamily that regulates AKT phosphorylation in C4-2 prostate cancer cells. In support of our conclusion that inhibition of AKT phosphorylation by class I/II HDACIs is unlikely mediated by their genomic effect in the nucleus, HDAC3 is one of the few class I/II HDAC proteins that are localized in the cytoplasm, plasma membrane, and the nucleus (McKinsey *et al*, 2000; Yang *et al*, 2002).

Aberrantly activated AKT has long been recognized as an attractive therapeutic target. AKT inhibitors are currently being utilized for clinical trials. In most scenarios, they exhibit optimal anti-cancer activities only when combined with the other therapeutic agents (Hirai *et al*, 2010; Pei *et al*, 2016). Particularly in prostate cancer, one of the major hurdles is the existence of negative feedback between AKT and AR signaling pathways, making a single blockage of AKT signaling inefficient in the treatment of prostate cancer. Therefore, based on the findings in this study and others, there are several reasons that inhibition of HDAC3 might be an ideal target for the treatment of prostate cancer.

**Figure 8.  The HDAC3 inhibitor suppresses SPOP-mutated prostate cancer cell growth.**

A       22Rv1 cells stably infected with SPOP F133V mutant lentivirus were treated with 3 μM of RGFP966 for 24 h and harvested for Western blot with the indicated antibodies.

B, C    Representative images of 3D Matrigel cultures 22Rv1 cells stably infected control or SPOP F133V lentivirus at day 5 post-treatment of RGF966 are shown in (B) with the quantitative data in a box plot (C). The lentivirus transfection efficiency is at least 95% of total cells. In graph (C), the description for box plot is the same as the figure legend in Fig 7H. Data are shown as means ± SD. DMSO-Lenti-EV (*n* = 188) versus RGFP966-Lenti-EV (*n* = 139): ***P = 3.05e-11, DMSO-SPOP-F1333V (*n* = 130) versus RGFP966-SPOP-F133V (*n* = 193): ***P = 2.85e-12 were performed by Wilcoxon rank sum test with continuity correction. Scale bar, 100 μm.

D–F     Representatives of patient-derived SPOP-mutant organoids at day 5 post-treatment of DMSO or 3 μM of RGF966 are shown in (D) with the quantitative data of OD value at 490 nm in (E). ***P = 1.43e-04 was performed using the unpaired two-tailed Student's *t*-test. Based on the observed growth rate of untreated SPOP-mutated organoids, greater than 50% of organoids reach 10 μm in diameter at day 5. "10 μm" was set as the cutoff value. The number of organoids with the diameter > 10 μm) from at least five fields were counted and analyzed (F). Data are shown as means ± SEM. ***P = 3.36e-05 was performed by the unpaired two-tailed Student's *t*-test. Scale bar, 100 μm.

G, H    Patient-derived SPOP-mutant organoids were treated with vehicle (DMSO) or 3 μM of RGF966 for 5 days and harvested for IFC for phosphorylated AKT (S473) (G) and AR (H). Scale bars, 50 μm.

Source data are available online for this figure.

Firstly, a previous study has revealed that increased expression of HDACs and activation of PI3K/AKT-mTORC1 pathway commonly occur in prostate cancer (Ellis *et al*, 2013), suggesting that both are highly relevant to prostate cancer. In this study, we reveal a positive correlation between HDAC3 expression and AKT phosphorylation at the molecular level, a spontaneous prostate cancer mouse model, tumor-derived organoids, and patient specimens. Indeed, one recently published paper demonstrated that only a combined treatment with pan class I/II HDACIs (e.g., LBH-589) and PI3Ki (BKM-120) can effectively inhibit the growth of aggressive MYC-driven medulloblastoma in mice (Tzivion *et al*, 2011). Mechanically, it has been shown that pan HDACIs upregulate FoxO1 mRNA and protein levels, while PI3Ki induces nuclear accumulation of FoxO1 protein. Different from the strategy used for brain tumor treatment, we demonstrated in the present study that both genetic deletion of or pharmacological inhibition of HDAC3 alone is sufficient to block prostate cancer cell growth *in vitro* and *in vivo*, highlighting the value of single-agent targeting of HDAC3 for prostate cancer treatment.

Secondly, HDACIs have been associated with the interruption of the AR pathway (Welsbie *et al*, 2009), which makes HDACIs somehow efficient for the treatment of prostate cancer. Given that AKT signaling and AR are two critical pathways in driving the pathogenesis of prostate cancer, the synergetic inhibition of these two pathways might pose a huge threat on the progression of prostate cancer. Indeed, the pharmacological inhibition of both pathways has led to a complete prostate tumor regression in a *Pten*-deficient murine prostate cancer model and in human prostate cancer xenografts (Carver *et al*, 2011). Interestingly, unlike the FDA-approved antiandrogen enzalutamide, which suppresses AR signaling pathway while inducing *AR* mRNA expression, we observed that HDAC3-specific inhibitor RGFP966 blocked expression of both *AR* mRNA and its downstream target genes including *NKX3.1*, *PSA*, and *TMPRSS2* (Fig EV5D). Most strikingly, RGFP966 inhibits the expression of both full-length and variant AR proteins. Our data suggest that dual inhibition of AKT and AR signaling in prostate cancer, especially in those with genetic alterations such as *PTEN* loss, can be achieved by single administration of HDAC3-specific inhibitor.

Thirdly, *SPOP* is the most frequently mutated gene in primary prostate cancer (Barbieri *et al*, 2012; Cancer Genome Atlas Research Network, 2015). Intriguingly, all the mutations in *SPOP* gene detected in prostate cancer thus far affect the binding of SPOP with its substrates. Previous biochemical studies identified that AR and transcription co-regulators including SRC-3, TRIM24, ERG, and BRD4 are degradation targets of SPOP and that mutations in SPOP result in aberrant elevation of these proteins and increased activities of AR (Geng *et al*, 2013, 2014; An *et al*, 2014, 2015; Theurillat *et al*, 2014; Groner *et al*, 2016; Blattner *et al*, 2017; Zhang *et al*, 2017). Notably, ectopic expression of F133V (a hotspot mutation of SPOP) in human prostate cancer cells or knock-in of this mutant in the mouse prostate results in an abnormal activation of AKT-mTORC1 signaling (Blattner *et al*, 2017; Zhang *et al*, 2017). Thus, *SPOP* mutations uncouple the negative feedback between AKT and AR signaling pathways and drive prostate tumorigenesis and progression (Blattner *et al*, 2017). We provided evidence that treatment with the HDAC3-specific inhibitor RGFP966 not only effectively decreased AKT activities and the expression of full-length AR and splicing variants in SPOP-mutant-expressing prostate cancer cell lines and

patient-derived organoids, but also inhibited the growth of SPOP-mutant prostate cancer cells in 2D and 3D cultures. These findings stress that HDAC3 is also a viable therapeutic target for SPOP-mutated prostate cancer.

In summary, our findings have demonstrated for the first time that the cytoplasmic (non-genomic) activity of HDAC3 is required for AKT phosphorylation and AR upregulation in prostate cancer cells. Treatment of prostate cancer cells with HDAC3-specific inhibitor not only inhibits AKT-mTORC1 signaling, but also suppresses expression of AR and its downstream target genes. In addition, the promising findings in the pre-clinical models (Matthews *et al*, 2015; Jiang *et al*, 2017) suggest that HDAC3 inhibition might lead to appropriate cancer trials in the future. Given that both AKT and AR signaling are aberrantly activated in prostate cancers, especially those with PTEN deletions or SPOP mutations, dual inhibition of AKT and AR pathways by administrating the single-agent HDAC3 inhibitor makes HDAC3 inhibition an attractive strategy for prostate cancer treatment in the clinic.

# Materials and Methods

### Plasmids and reagents

The mammalian expression vectors for HA-tagged ubiquitin (HA-Ub), Myc-AKT, Flag-HDAC3 were all purchased from Addgene. HDAC3 and AKT mutant expression vectors were generated using the KOD-Plus Mutagenesis Kit (Toyobo). Insulin-growth factor 1 (IGF-1) and epidermal growth factor (EGF), cycloheximide (CHX) and HDAC inhibitors, including trichostatin A (TSA), suberoylanilide hydroxamic acid (SAHA), LBH589, GDC0068, RGFP966, and tubastatin A (Tuba) were purchased from Sigma-Aldrich. The antibodies used are as follows: Anti-ERK 2 (D-2) (sc-1647), AKT (sc-5298), AR (H-280) (sc-13062), and anti-Myc tag from Santa Cruz Biotechnology; Anti-HA from Covance; Flag-M2 (F-3165) from Sigma; Cleaved caspase-3 (Asp175) (D3E9) (9579S), Tubulin (9F3) (2128S), 4E-BP1 (53H11) (9644S), Phospho-4E-BP1 (2855S), P70 S6 Kinase (9202S), Phospho-S6K (T389) (4858S), Phospho-AKT-Thr308 (9275S), Phospho-AKT-Ser473 (9271S) and PTEN (9559S), total AKT antibody (9272S, recognizing three AKT isoforms) from Cell Signaling Technology; HDAC3 (ab16047), FBP1 (ab109732), and Ki67 (ab15580) from Abcam. Anti-acetyl-lysine antibody from Upstate; E-cadherin (610181) from BD Biosciences; SPOP (16750-1-AP) from Proteintech Group Inc. For Western blots, all the antibodies were diluted 1 in 1,000, whereas for IFC, the antibodies were diluted 1 in 500. Matrigel Basement membrane Matrix (# 354248) was purchased from Corning Life Sciences. The secondary florescence antibodies (Alexa Fluor 488 and Alexa Fluor 594) were purchased from Thermo Fisher.

### Cell lines, cell culture, and 3D culture

The immortalized human embryonic kidney cell line 293T was purchased from ATCC (Manassas, VA) and cultured in Dulbecco's modified Eagle's medium (Invitrogen) supplemented with 10% FBS. The prostate cancer cell lines PC-3, 22Rv1, VCaP, and LNCaP were purchased from ATCC (Manassas, VA). C4-2 cells were acquired from UroCorporation. All prostate cancer cells applied in this study

were cultured in 10% FBS (Hyclone) in RPMI 1640 medium (Invitrogen) supplemented with penicillin and streptomycin. The cultured cells were maintained in a 37°C humidified incubator supplied with 5% $CO_2$.

For three-dimensional (3D) cultures, $2 \times 10^4$ of C4-2 or 22Rv1 cells were resuspended in 250 μl plain medium and seeded on the top of a thin layer of Matrigel in a 24-well plate. After 30 min, when the cells were settled down, they were covered with another layer of 10% Matrigel diluted with DMEM/F12 medium. The medium was changed with 500 μl of fresh and warm DMEM/F12 plus 5% FBS medium every 2–3 days.

## Real-time PCR

Total RNAs were extracted with TRIzol (Invitrogen) and reverse transcribed into cDNA with SuperScript III First-Strand Synthesis System (Promega). Quantitative PCR was done in the iQ thermal cycler (Bio-Rad) using the iQ SYBR Green Supermix (Bio-Rad) and in triplicate. The ΔCT was calculated by normalizing the threshold difference of certain gene with glyceraldehyde-3-phosphate dehydrogenase (GAPDH).

## Mouse maintenance and genotyping

*Hdac3* conditional knockout (*Hdac3$^{L/L}$*) mice were reported previously (Bhaskara *et al*, 2008) and kindly provided by Dr. Jennifer Westendorf at Mayo Clinic. *Pten* conditional knockout (*Pten$^{L/L}$*) mice were originally generated in the laboratory of Dr. Hong Wu at University of California Los Angeles (Wang *et al*, 2003) and purchased from the Jackson Laboratory. The *Pb-Cre*4 transgenic mice were generated originally in the laboratory of Dr. Pradip Roy-Burman, at the University of Southern California (Wu *et al*, 2001) and acquired from the National Cancer Institute (NCI) Mouse Repository. The cohorts of *Hdac3$^{pc-/-}$*; *Pten$^{pc-/-}$*, *Hdac3$^{L/L}$*; *Cre$^-$*;*Pten$^{L/L}$*, *Hdac3$^{pc-/-}$* ("wild-type"), and *Pten$^{pc-/-}$* male mice were generated from *Hdac3$^{pc+/-}$*; *Pten$^{pc+/-}$* males and *Hdac3$^{L/+}$*; *Pten$^{L/+}$* females, which were obtained by crossbreeding *Pb-Cre*4 males with *Hdac3$^{L/L}$* and *Pten$^{L/L}$* females. All mice were maintained under standard conditions of feeding, light, and temperature with free access to food and water. All experimental protocols were approved by the Institutional Animal Care and Use Committee (IACUC) at Mayo Clinic. Genotyping of wild-type and conditional alleles of *Pten* genes, as well as the *Cre* transgene, was performed according to previously described PCR protocols (Wu *et al*, 2001; Lesche *et al*, 2002). The *Hdac3* genotyping primer sequences are provided in Appendix Table S1.

## Transfection of expression vectors, RNA interference (siRNA), and shRNA

Transfections were performed either by electroporation using an Electro Square Porator ECM 830 (BTX) (Chen *et al*, 2010) or by Lipofectamine 2000 (Invitrogen). Approximately 75–90% transfection efficiencies were routinely achieved. The siRNA constructs were purchased from GE Dharmacon. The transfection for siRNA was performed using Lipofectamine® RNAiMAX (Thermo Fisher) according to the manufacturer's instruction. The siRNA sequence information is provided in Appendix Table S2. All shRNA constructs were purchased from Sigma and transfected with lentivirus.

## Western blot and immunoprecipitation (IP)

Western blotting was performed as described previously (Chen *et al*, 2010). Antibodies used for Western blotting were diluted at 1:1,000 to 1:2,000. The IP was carried out using an IP kit (Roche Applied Science) as described previously (Huang *et al*, 2006).

## Immunofluorescent cytochemistry and immunohistochemistry (IHC)

Immunohistochemistry and IFC were performed as previously described (Xu *et al*, 2014). The sections for IHC were cut at 4 μm thickness. Antigen retrieval was conducted via heat-induced epitope retrieval, with 10 mM sodium citrate buffer (pH 6.0) for all antibodies used in this study. Antibodies for HDAC3, PTEN, and P-AKT (Ser473) were incubated at 4°C overnight. Color was developed with SignalStain® DAB Substrate Kit.

## Microscopic observations and analysis

Immunohistochemistry and H&E staining were observed with a Leica light microscope (10×, 20×, and 40×). Cell growth of 3D Matrigel culture was recorded using Leica microscope at day 7, and the diameter of 3D culture was analyzed with Leica software—LAS EZ. IFC staining images were obtained via a Zeiss LSM 780 confocal microscope. Staining intensity and staining percentage for each tissue were graded using set criteria. Staining intensity was graded into four categories: 0, 1, 2, and 3. Specifically, 0 represents no staining, 1 low staining (staining obvious only at ×400), 2 medium staining (staining obvious at ×100 but not ×40), and 3 strong staining (staining obvious at ×40). Stain percentage was graded 1 for 0–33% positive cells, 2 for 34–66%, and 3 for 67–100%. The final SI score for each staining was obtained by multiplying values obtained from staining percentage and intensity and used for correlation analysis.

## Clonogenic survival

The procedure was conducted by following a previous report (Franken *et al*, 2006). Briefly, an appropriate number of cells for different dosages of drugs were plated onto 6-well plate. At the following day, cells were treated with DMSO, GDC0068, RGFP966, and the combination of GDC0068 and RGFP966. Four days post-treatment, cells were cultured with fresh medium without drugs for another 8 days. Around 12 days later, colonies were fixed with acetic acid: methanol (1:7) for 30 min and stained with (0.5% w/v) crystal violet for 1 h. Colonies were gently washed with running tap water. Colonies with more than 50 cells were counted, and the number of colonies was normalized to untreated group. The linear regression was applied to generate the survival curve.

## Patient-derived organoid culture

A PTEN-deleted and a SPOP-mutant (G131R) organoid lines were kindly provided from Dr. Chen laboratory in Memorial Sloan-Kettering Cancer. The detailed extraction and culture procedures were referred to two published papers (Gao *et al*, 2014; Drost *et al*, 2016). Briefly, the organoids were seeded onto a thin layer of Matrigel and passaged every 3–4 days. The recipe of culture medium can be seen in the previous report (Drost *et al*, 2016).

**The paper explained**

**Problem**
Prostate cancer is one of the leading causes of cancer death in American and European men. Both AKT-mTOR and androgen receptor (AR) signaling pathways are aberrantly activated in prostate cancer due to PTEN deletion or SPOP mutation, two genetic lesions occurring in up to 80% of advanced prostate cancer. However, the existence of negative feedback regulation between AKT and AR signaling pathways makes the single treatment with AKT inhibitor or AR inhibitor somehow inefficient.

**Results**
The major findings from the present study include the following: (i) HDAC3 is the only class I/II HDAC protein that regulates AKT phosphorylation; (ii) overexpression of HDAC3 correlates with increased AKT phosphorylation in prostate cancer patient specimens; (iii) genetic deletion of HDAC3 suppresses prostate tumorigenesis and progression by concomitant blockade of AKT-mTOR and AR signaling in PTEN knockout prostate tumors; (iv) this effect of HDAC3 is mediated by its function in the cytoplasm; (v) pharmacological inhibition of HDAC3 using a selective HDAC3 inhibitor RGFP966 inhibits growth of both PTEN-deficient and SPOP-mutated prostate cancer cells in culture, patient-derived organoids and xenografts in mice.

**Impact**
Our findings provide mechanistic explanation as to how AKT phosphorylation and AR expression is regulated by HDAC3 and highlight that dual inhibition of AKT-mTOR and AR by a single agent is clinically achievable by using small molecule inhibitors of HDAC3 in prostate cancer, especially those with PTEN deletion or SPOP mutation.

Informed consent was obtained from all subjects and that the experiments conformed to the principles set out in the WMA Declaration of Helsinki and the Department of Health and Human Services Belmont Report.

**Mouse xenograft and tumor analysis**

The 6-week-old NOD-SCID IL-2-receptor gamma null (NSG) male mice were generated in house and randomly divided into different experimental groups as indicated. Mice were injected with $1 \times 10^7$ of C4-2 cells in 100 μl of Matrigel matrix (BD Bioscience) in the left flank. GDC-0068 was formulated in 0.5% methylcellulose/0.2% Tween-80 (MCT) and administered via oral gavage daily at 50 mg/kg (Lin *et al*, 2013), while RGFP966 was dissolved in DMSO and diluted in a vehicle of 30% (wt/vol) hydroxypropyl-β-cyclodextrin and 100 mM sodium acetate (pH 5.4) as reported previously (Malvaez *et al*, 2013) and administrated subcutaneous injection (s.c) at 25 mg/kg (L) or 50 mg/kg (H). The drug administration was 5 days a week for 21 consecutive days. After implantation of tumor cells into mice, tumors were monitored until they reached mean tumor volumes of 180–350 mm³ and distributed into four groups (7 mice/group). Tumor growth was measured externally by caliper twice a week. The protocol for conducting this mouse xenograft experiment was approved by Mayo Clinic IACUC.

**Generation of graphs and statistical analysis**

Graphs were generated by using Graphpad Prism 5 project (Graphpad Software Inc, CA, USA) or Microsoft Office Excel 2010. All numerical data are presented as mean ± SEM or mean ± SD as required. The survival percentage was compared by Chi-square tests. Differences between groups were compared by unpaired *t*-tests or Wilcoxon rank sum test with continuity correction by R software version 2.15.0 (http://www.r-project.org). The following symbols were used to denote statistical significance: *$P < 0.05$, **$P < 0.01$, ***$P < 0.001$.

**Expanded View** for this article is available online.

## Acknowledgements
This work was supported in part by grants from NIH (CA134514, CA130908, and CA193239 to H.H.), U.S. Department of Defense (W81XWH-14-1-0486 to H.H.), the National Natural Science Foundation of China (31560320 to D.W.), and the Natural Science Foundation of Jiangxi Province (20142BAB215049 to D.W.).

## Author contributions
HH, DWa, and RZ conceived the study. YYan, JA, YYang, DWu, YB, WC, LM, JC, ZY, YH, XJ, YP, SW, XH, SJW, RJK, JJW, YC, and DWa generated organoid and mouse models, performed experiments, and analyzed the data. JZ, TM, and LW supervised histological and IHC data analysis. HH, YYan, WX, RZ, and DWa wrote the manuscript.

## Conflict of interest
The authors declare that they have no conflict of interest.

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
