## [Review Process File · EMBO Molecular Medicine]

Dual inhibition of AKT-mTOR and AR signaling by targeting HDAC3 in *PTEN*- or *SPOP*-mutated prostate cancer

Yuqian Yan, Jian An, Yinhui Yang, Di Wu, Yang Bai, William Ca, Linlin Ma, Junhui Chen, Zhendong Yu, Yundong He, Xin Jin, Yunqian Pan, Tao Ma, Shangqian Wang, Xiaonan Hou, S. John Weroha, R. Jeffrey Karnes, Jun Zhang, Jennifer J. Westendorf, Ligu Wang, Yu Chen, Wanhai Xu, Runzhi Zhu, Dejie Wang, and Haojie Huang

Review timeline:

Submission date:	11 September 2017
Editorial Decision:	9 October 2017
Revision received:	10 January 2018
Editorial Decision:	26 January 2018
Revision received:	7 February 2018
Accepted:	8 February 2018

Editors: Roberto Buccione and Céline Carret

Transaction Report:

1st Editorial Decision

9 October 2017

Thank you for the submission of your manuscript to EMBO Molecular Medicine. We have now heard back from the three referees whom we asked to evaluate your manuscript.

As you will see from the supportive and rather overlapping comments below, the three referees are enthusiastic about the study and suggest a few recommendations to further improve conclusiveness by performing additional analyses and experiments, but also by providing better descriptions and clarifications here and there. More insights into the mode of action would also be desirable (see ref2).

I look forward to receiving your revised manuscript.

***** Reviewer's comments *****

Referee #1 (Comments on Novelty/Model System for Author):

I wonder how specific HDAC3 inhibitors will be in clinical practice and how well will they be tolerated given HDAC3 knockout is lethal.

Referee #1 (Remarks for Author):

In the manuscript "Dual inhibition of AKT-mTOR and AR signaling by targeting HDAC3 in *PTEN*- or *SPOP*-mutated prostate cancer" the authors have demonstrated the role of HDAC3 in regulating AKT activation in response to growth factors, and AR activation. The authors suggest that HDAC3 inhibitors can be used for the dual inhibition of the both pathways

especially in the context of PTEN/SPOP loss that show increased AKT activity. The authors identify an AKT binding domain (ABD) that is required for the interaction of HDAC3 with AKT and its subsequent regulation. They also show the regulation of the AR signalling by HDAC3 by measuring level of AR and its target genes. Using pre-clinical/cell culture based models they further showed HDAC3 inhibition represses PTEN loss-induced AKT activation and tumour growth.

They have conducted in depth analysis of the major findings of this paper using robust experimental controls and the conclusions of the study are supported by highly competent experimental strategy.

Clearly, from the clinical point of view, HDAC3 inhibition would be a breakthrough for the management of PTEN/SPOP null prostate tumours.

Major comment:

As authors themselves pointed out in reference #16 and #36 that similar observations have been made earlier and HDAC3 is known to regulate both AR and downstream pathways, and AKT respectively. Authors should comment on the novelty of this study in order to justify and highlight how does this study represent a scientific advance.

Other issues:

1. In figure 1A authors have used PAN HDAC inhibitor(s). They should have used RGFP966 which they say is a specific inhibitor which should have been in the first place. Does this inhibitor affect AKT ubiquitination, authors should comment on this?
2. While in figure 1C authors show marked effect of HDAC3 knockdown on T308 phosphorylation and less so on S473 phosphorylation. But in figure 1D they chose to test the S473 phosphorylation using IHC on TMAs to corroborate this data, which is less convincing. Is there a validated antibody for IHC application to detect T308? This should be used here to correlate in vitro findings.
3. Figure 4A: Authors show colocalisation of HDAC3 and AKT, this should be quantified to determine what fraction of HDAC3 colocalises with AKT on plasma membrane?
4. Authors found the requirement of ABD in HDAC3 in its interaction with AKT, was there a consensus sequence found for AKT interaction? This should be discussed.

Referee #2 (Comments on Novelty/Model System for Author):

While the Hdac3:AKT connection has been published, the links in prostate cancer are novel.

Referee #2 (Remarks for Author):

Yan et al. report a cytoplasmic/plasmic membrane role for HDAC3 in the regulation of AKT signaling in prostate cancer cells. Specifically, they suggest that HDAC3-mediated deacetylation of Lys20 and Lys14 of AKT is required for AKT polyubiquitination, localization at the cell membrane and subsequent phosphorylation. Such a mechanism could have therapeutic implications for prostate cancer, given the frequent activation of PI3K-AKT signaling in this disease (commonly through loss of the PTEN or SPOP tumor suppressor genes).

Yan et al. go on to show that Hdac3 deletion extends survival in a PTEN-driven mouse model of prostate cancer and that treatment of prostate cancer cells with an HDAC3-selective inhibitor, RGFP966, impairs 3D growth as well as xenograft growth in immunocompromised mice. Of particular clinical interest is both the reduction in AKT signaling as well as the loss of androgen receptor expression following HDAC3 inhibition. However, while therapeutically desirable, these pleiotropic effects make mechanistic conclusions about the specific role of HDAC3 in the regulation of PI3K-AKT signaling and how the inhibition of this pathway contributes to the efficacy of HDAC3 inhibitors less clear.

Specific Comments:

1. HDAC3 is only active in a complex with NCOR or SMRT, but these factors were not considered in this work. It is commonly accepted that HDAC3 is not the rate-limiting component of this complex, so modulating its levels may not change the activity of this complex. In Fig. 2A very modest changes in HDAC3 levels appeared to have dramatic effects on AKT acetylation levels, so it would be wise to assess NCOR/SMRT levels to determine if HDAC3 is rate-limiting in this situation. The stoichiometry of the complexes should be carefully considered, as it is possible that APPL1 binding would prevent NCOR/SMRT from activating the complex.

2. CHX treatment (Fig. 1A) suggests that the effects of HDACi on AKT phosphorylation occur independently of HDACi-induced changes in transcription and subsequent protein synthesis. As such, one would expect direct effects of HDAC3-inhibition (with RGFP966) on p-AKT levels to require only 20-30 min to manifest themselves, yet p-AKT levels are not assayed until 24 hours after RGFP966 treatment. Such a long treatment time could cloud interpretation as others have shown a significant induction of growth arrest and/or apoptosis in sensitive cell lines following 24 hours of RGFP966 treatment. Thus, earlier time points following HDAC3 inhibition should be examined in a time course experiment.

3. The model that HDAC3 regulates AKT localization at the cell membrane is an interesting one, however the immunofluorescence in Fig. 4 lacks a cell membrane marker such as E-cadherin that is necessary to more confidently conclude that these proteins are indeed colocalizing there. In addition, the IF for HDAC3 looks unusual. While there may well be cytoplasmic targets of HDAC3 activity, HDAC3 should still be enriched in the nucleus, but this does not appear to be the case. Either these PCa cells are relatively unique in their localization of HDAC3 (which is not supported by the cell fractionation of over expressed HDAC3) or the IF is not working well. Cell fractionation should be performed on endogenous HDAC3 in cell lines to look at the localization of the proteins (rather than merely overexpression systems in which the abundance of protein could potentially affect localization).

4. Its unclear what the baseline prostate phenotype is for the conditional Hdac3 deletion. This is a critical control given the effects on AR function. A full characterization is needed to understand how the loss of PTEN affects the phenotype. From the H&E in Fig. 6F, there could be a loss in the cellularity of the prostate that could account for the slowing of tumorigenesis.

5. In spite of the thorough biochemical characterization of the effect of HDAC3 loss or overexpression on AKT phosphorylation, ultimately PTEN loss and elevated AKT phosphorylation did not predict an increased sensitivity to HDAC3 inhibition- at least not in the two cell lines examined (Fig. 7C). In addition, the loss of Androgen receptor upon HDAC3i was even more dramatic than loss of p-AKT (Fig. 7F). Therefore, efforts should be made to clarify the contribution of PI3K-AKT inhibition vs. AR reduction to the overall efficacy of this drug. For instance, PC3 cells do not express AR, yet have lost PTEN- are they similarly sensitive to RGFP966. Or- how would exogenous expression of AR or myr-Akt (which should no longer require HDAC3 to localize to membrane) affect the IC50? In addition, is the PTEN tumor model dependent on AR and how does Hdac3 deletion affect AR levels in these tumors?

Minor Comments:

1. For co-IPs, inputs should be blotted for all transfected proteins, particularly if conclusions are being drawn about relative binding (ex. Fig. 4D)
2. IHC and IF should not be described as quantitative
3. In figure 2B, it appears that flag-Hdac3 is present in all lanes? In addition the increase in AKT ubiquitination is substantial, yet p-AKT levels are not increased- is this an outlier?
4. What information is provided in Fig. 2F that is not already provided in Fig. 2D?
5. p-AKT blot in Fig. 6c is very over-exposed.
6. why are WT and Hdac3^{-/-} missing from 6c and 6e?
7. review figures for typos (e.g. Fig 6I "fields") and mislabeling (e.g. Fig. 4e- HA-Hdac3 transfected, yet IB says Flag)
8. Some experiments contain multiple proteins that contain the same epitope tag, which is quite confusing.

Referee #3 (Comments on Novelty/Model System for Author):

The authors use a combination of genetically modified mouse models, organoids, and PDX as well as cell and molecular biology to address whether HDAC3 affects AKT activity in prostate cancer.

Referee #3 (Remarks for Author):

This is a beautiful piece of research that moves the field forward. I have only a few questions none of which should require additional experimentation. The authors may insert a sentence or two into the paper in response to some of these as they see fit.

p. 6 Figure 1C shows that HDAC3 is the only HDAC required for AKT phosphorylation in C4-2 cells. But this is not necessarily so in all prostate cancer cells. A change in wording please.

p. 7-8 Figure 2 establishes that HDAC3 and growth factors affect the acetylation and polyubiquitination of AKT. Three forms of Akt, Akt1, 2, and 3 play distinctive roles in signaling and cancer. Do the authors know whether the effects of growth factors and HDAC3 relate to any of particular form of AKT?

p. 9 Does the ABD deletion mutant of HDAC3 affect gene expression? Are the domains of HDAC3 important to gene expression and AKT functions common or distinct?

p. 10 The authors show that in prostate cancer cells cytoplasmic HDAC3 regulates Akt. Another group found that in leukemia, nuclear interactions of these proteins may occur. Do the authors feel that they should indicate that cell or cancer type dependencies may exist? My suspicion is that cytoplasmic interaction may be most important to Akt phosphorylation but why not leave the door open a bit on this issue?

Can the authors suggest what mediates increased cytoplasmic rather than nuclear HDAC3 activity in prostate cancer with PTEN deletions or SPOP mutations?

p. 16 How significant are the cell type differences in the clonogenic assays (Fig. 7B, C)?

p. 17 Inhibitors of mTOR may have different effects on S6 and EIF 4EBp1. Do the authors have any data or thoughts on the latter mTOR output?

P18. How effective was the viral transduction of the spheres in figure 8B?

p. 19 In studies with organoids the authors report a decrease in organoids with a diameter greater than 10 μ M. Yet I suspect that there is heterogeneity in the responses of individual organoids to drugs. When spheres are formed in low adherence plates how diverse is the size of the spheres that form? And despite the fact that the organoids (are these really organoids as they come from a cell line?) are genetically homogeneous their response is heterogeneous. Why?

The authors should describe how they isolated and cultured the patient-derived organoids in more detail.

The authors should explain why they set "30 μ m in diameter" as the cut-off value of the growing organoids and how they calculated the percentage of that.

Is HDAC H134Q, a deacetylase inactivation mutant, common in prostate cancer? What is the frequency of this mutation and is there any relationship between prognosis and the frequency of this mutation?

On page 27 of the methods, what is the criterion for establishing the intensity grade of staining? How did the authors divide "low staining" and "strong staining"?

Referee #1 (Comments on Novelty/Model System for Author):

I wonder how specific HDAC3 inhibitors will be in clinical practice and how well will they be tolerated given HDAC3 knockout is lethal.

Reply: We fully agree with the Reviewer that HDAC3 is important for embryogenesis. Based upon our findings and those in the literature, in adult tissues it is still clinically practical to target overexpressed HDAC3 for cancer treatment due to the finding that HDAC3 is overexpressed in tumors compared to normal tissues and overexpressed HDAC3 plays an essential role in oncogenesis - another potential scenario of “oncogene addiction”.

Referee #1 (Remarks for Author):

Major comment:

As authors themselves pointed out in reference #16 and #36 that similar observations have been made earlier and HDAC3 is known to regulate both AR and downstream pathways, and AKT respectively. Authors should comment on the novelty of this study in order to justify and highlight how does this study represent a scientific advance.

Reply: That’s an excellent point. As to the regulation of AKT by HDAC3, Long and colleagues claim that HDAC3 and AKT are co-localized exclusively in the nucleus where they believe AKT deacetylation and phosphorylation occur (Long, J., et al, *Leukemia* 31(12): 2761-70, 2017). However, we and the others show that HDAC3 harbors a nuclear export sequence that is important for its exportation to cytoplasm (Longworth, M.S. and L.A.Laimins, *Oncogene* 25(32): 4495-500, 2006). We demonstrate for the first time that the cytoplasm form of HDAC3 is as sufficient as the wild-type counterpart to promote AKT phosphorylation (Fig 4C and EV2C-E). We also demonstrate for the first time that both K14 and K20 residues on AKT are critical for polyubiquitination, phosphorylation, and activation of AKT by HDAC3 (Fig 4F and 4G).

As to the regulation of AR by HDAC3, it has been shown previously that expression of AR target genes is reduced by HDAC3 knockdown, although the underlying mechanisms remain poorly understood. We demonstrate in the current study that the AR regulatory function of HDAC3 is mainly mediated by its activity in the cytoplasm (Fig EV4E-G), a novel contribution to the field.

Other issues:

1. In figure 1A authors have used PAN HDAC inhibitor(s). They should have used RGFP966 which they say is a specific inhibitor which should have been in the first place. Does this inhibitor affect AKT ubiquitination, authors should comment on this?

Reply: We have performed two new experiments to address these concerns. We demonstrated that: 1) RGFP966 inhibits AKT phosphorylation in a time-dependent manner (Fig 1D); 2) RGFP966 inhibits AKT ubiquitination (Fig 2F).

2. While in figure 1C authors show marked effect of HDAC3 knockdown on T308 phosphorylation and less so on S473 phosphorylation. But in figure 1D they chose to test the S473 phosphorylation using IHC on TMAs to corroborate this data, which is less convincing. Is there a validated antibody for IHC application to detect T308? This should be used here to correlate in vitro findings.

Reply: This is an excellent point. We consistently found in the lab that the affinity of AKT S473 phospho antibody is much better than the T308 phospho antibody, which is also reflected in the western blots (WB) shown in Fig 1C that the intensity of WB bands for S473 phosphorylation was much stronger than that of T308 phosphorylation, even though the same amount of proteins was loaded in each lane in each WB experiment with the same running and transfer conditions as well as the exposure time of WB. However, by quantifying the WB data, we demonstrated that the degree of S473 phosphorylation downregulation induced by HDAC3 knockdown was similar to that of T308 phosphorylation downregulation (Fig 1C). Thus, the level of S473 phosphorylation does reflect the effect of HDAC3 on AKT phosphorylation, and S473 IHC data from TMA is representative.

3. Figure 4A: Authors show colocalisation of HDAC3 and AKT, this should be quantified to determine what fraction of HDAC3 colocalises with AKT on plasma membrane?

Reply: We calculated the percentage of cells with co-localization of AKT and HDAC3 in at least 5 fields. Approximately 80% of LNCaP cells and 85% of C4-2 cells showed the co-localization of these two proteins on plasma membrane. We have provided this information in the legend of Fig 4A.

4. Authors found the requirement of ABD in HDAC3 in its interaction with AKT, was there a consensus sequence found for AKT interaction? This should be discussed.

Reply: This is an excellent point. To address this concern, we examined more than 100 AKT-interacting proteins. We noticed that protein sequences in the AKT-binding region are reported only in five of them (APPL1, YB-1, BRCA1, MEN1 and DAB2), but we found no consensus AKT-binding sequence between these five proteins and HDAC3, suggesting that the ABD in HDAC3 is unique. We have included this discussion in the manuscript on page 9.

Referee #2 (Comments on Novelty/Model System for Author):

While the Hdac3:AKT connection has been published, the links in prostate cancer are novel.

Reply: We thank the Reviewer for the positive comment on the novelty of our manuscript.

Referee #2 (Remarks for Author):

Yan et al. report a cytoplasmic/plasmic membrane role for HDAC3 in the regulation of AKT signaling in prostate cancer cells. Specifically, they suggest that HDAC3-mediated deacetylation of Lys20 and Lys14 of AKT is required for AKT polyubiquitination, localization at the cell membrane and subsequent phosphorylation. Such a mechanism could have therapeutic implications for prostate cancer, given the frequent activation of PI3K-AKT signaling in this disease (commonly through loss of the PTEN or SPOP tumor suppressor genes).

Yan et al. go on to show that Hdac3 deletion extends survival in a PTEN-driven mouse model of prostate cancer and that treatment of prostate cancer cells with an HDAC3-selective inhibitor, RGFP966, impairs 3D growth as well as xenograft growth in immunocompromised mice. Of particular clinical interest is both the reduction in AKT signaling as well as the loss of androgen receptor expression following HDAC3 inhibition. However, while therapeutically desirable, these pleiotropic effects make mechanistic conclusions about the specific role of HDAC3 in the regulation of PI3K-AKT signaling and how the inhibition of this pathway contributes to the efficacy of HDAC3 inhibitors less clear.

Specific Comments:

1. HDAC3 is only active in a complex with NCOR or SMRT, but these factors were not considered in this work. It is commonly accepted that HDAC3 is not the rate-limiting component of this complex, so modulating its levels may not change the activity of this complex. In Fig. 2A very modest changes in HDAC3 levels appeared to have dramatic effects on AKT acetylation levels, so it would be wise to assess NCOR/SMRT levels to determine if HDAC3 is rate-limiting in this situation. The stoichiometry of the complexes should be carefully considered, as it is possible that APPL1 binding would prevent NCOR/SMRT from activating the complex.

Reply: This is an excellent point. We agree that the level of transfected Flag-HDAC3 in Fig 2A appeared to be very low, but we believe this is a western blot film exposure issue. To address this issue, we performed additional western blots using both anti-Flag and anti-HDAC3 antibodies. The new results showed that a substantial increase in HDAC3 protein level was achieved by transfecting Flag-tagged HDAC3 in these experiments (Fig 2A).

To investigate whether APPL1 binding would potentially prevent NCOR/SMRT from activating the NCOR/SMRT/HDAC3 complex, we knocked down APPL1 and performed co-immunoprecipitation experiments. While APPL1 knockdown did not affect the expression levels of endogenous NCOR and SMRT and transfected HA-HDAC3, it did increase HDAC3 interaction with NCOR/SMRT (Fig EV2F). These new data suggest that APPL1 binding can prevent NCOR/SMRT from binding to HDAC3.

2. CHX treatment (Fig. 1A) suggests that the effects of HDACi on AKT phosphorylation occur independently of HDACi-induced changes in transcription and subsequent protein synthesis. As such, one would expect direct effects of HDAC3-inhibition (with RGFP966) on p-AKT levels to require only 20-30 min to manifest themselves, yet p-AKT levels are not assayed until 24 hours after RGFP966 treatment. Such a long treatment time could cloud interpretation as others have shown a significant induction of growth arrest and/or apoptosis in sensitive cell lines following 24 hours of RGFP966 treatment. Thus, earlier time points following HDAC3 inhibition should be examined in a time course experiment.

Reply: We agree with the Reviewer. We performed new experiments with earlier time points of RGFP966 treatment. We demonstrated that the effect of HDAC3 inhibition by RGFP966 on AKT phosphorylation occurred as early as 30 min post treatment. The new data provide further support to the notion that the effect of HDAC3 on AKT phosphorylation is likely an immediate event.

3. The model that HDAC3 regulates AKT localization at the cell membrane is an interesting one, however the immunofluorescence in Fig. 4 lacks a cell membrane marker such as E-cadherin that is necessary to more confidently conclude that these proteins are indeed colocalizing there. In addition, the IF for HDAC3 looks unusual. While there may well be cytoplasmic targets of HDAC3 activity, HDAC3 should still be enriched in the nucleus, but this does not appear to be the case. Either these PCa cells are relatively unique in their localization of HDAC3 (which is not supported by the cell fractionation of over expressed HDAC3) or the IF is not working well. Cell fractionation should be performed on endogenous HDAC3 in cell lines to look at the localization of the proteins (rather than merely overexpression systems in which the abundance of protein could potentially affect localization).

Reply: As suggested by the Reviewer, we performed new IFC experiments to examine the co-staining of AKT and HDAC3 with cell membrane E-cadherin. Our new data showed that both HDAC3 and AKT are colocalized very nicely with E-cadherin in majority of cells examined, providing further support to our conclusion that AKT and HDAC3 colocalize on the membrane.

We also agree that the images of C4-2 cells were not good representatives that were inconsistent with the fractionation results. We therefore repeated the experiments by using a newly acquired anti-HDAC3 antibody. Our new data showed that while HDAC3 protein can be detected in the plasma membrane and cytoplasm, a significant portion of them were localized in the nucleus (Fig 4A).

As suggested by the Reviewer, we performed new cell fractionation experiment by focusing on endogenous HDAC3 in LNCaP cells. Our new data showed that majority of AKT proteins interacted with HDAC3 in the cytoplasm, although a weaker AKT-HDAC3 interaction was detectable in the nucleus (Fig 4B).

4. It's unclear what the baseline prostate phenotype is for the conditional Hdac3 deletion. This is a critical control given the effects on AR function. A full characterization is needed to understand how the loss of PTEN affects the phenotype. From the H&E in Fig. 6F, there could be a loss in the cellularity of the prostate that could account for the slowing of tumorigenesis.

Reply: As indicated by morphological/histological (H&E) analysis (Fig 6F) and quantitative analysis of normal versus malignant acini, including low grade prostatic intraepithelial neoplasia (LGPIN), high grade PIN (HGPIN) and cancer (Fig 6G), conditional deletion of Hdac3 alone had no overt effect on the baseline phenotype of the prostate (Fig 6F and 6G). Our IHC analysis demonstrated that the negligible AKT phosphorylation (S473) in the Hdac3 knockout prostate was similar to that in the wild-type counterpart (Fig 6B-iii). Additionally, we showed that expression of AR protein and its target genes was downregulated in conditional Hdac3-deleted prostates compared to that in wild-type counterparts (Fig EV4C and 4D). Given the low levels of AKT phosphorylation

and AR activity in conditional Hdac3-deleted prostate, a scenario similar to that in wild-type prostate, it is not surprising that Hdac3 knockout alone did not result in any malignant changes in the prostate (Fig 6F and 6G).

To determine whether a loss in the cellularity of the prostate accounts for the slowing of Pten deficiency-induced tumorigenesis caused by Hdac3 knockout, we performed IHC analysis of cleaved caspase-3 as a surrogate of apoptosis. Our new data demonstrated that Hdac3 loss had no overt effect on apoptosis (Fig EV4A and 4B). Thus, the slowing of Pten deficiency-induced tumorigenesis (Fig 6F and 6G) caused by Hdac3 knockout is likely mediated by decreased cell proliferation (Fig 6H and 6I).

5. In spite of the thorough biochemical characterization of the effect of HDAC3 loss or overexpression on AKT phosphorylation, ultimately PTEN loss and elevated AKT phosphorylation did not predict an increased sensitivity to HDAC3 inhibition- at least not in the two cell lines examined (Fig. 7C). In addition, the loss of Androgen receptor upon HDAC3i was even more dramatic than loss of p-AKT (Fig. 7F). Therefore, efforts should be made to clarify the contribution of PI3K-AKT inhibition vs. AR reduction to the overall efficacy of this drug. For instance, PC3 cells do not express AR, yet have lost PTEN- are they similarly sensitive to RGFP966. Or- how would exogenous expression of AR or myr-Akt (which should no longer require HDAC3 to localize to membrane) affect the IC50? In addition, is the PTEN tumor model dependent on AR and how does Hdac3 deletion affect AR levels in these tumors?

Reply: As suggested by the Reviewer, we overexpressed AR in AR-negative, PTEN-null cell line PC-3. Consistent with the previous report (Lin et al., PNAS 98: 7200-5, 2001), overexpression of AR triggered apoptosis in PC-3 cells. We therefore cannot pursue further AR-related experiments using PC-3 because we concern the apoptotic effect of AR overexpression in PC-3 cells could potentially cloud the interpretation of the effect of AR on the drug sensitivity. Therefore, AR and constitutively active (myristoylated) AKT (CA-AKT) were exogenously expressed in C4-2 cells (PTEN-null, but AR-positive) and drug sensitivity was examined in these cells. Clonogenic survival assays showed that exogenous expression of AR or CA-AKT alone only partially blocked RGFP-induced inhibition of C4-2 cell growth (Fig EV5C and 5D), suggesting that both AKT inhibition and AR reduction are important for RGFP966 inhibition of cell growth.

As suggested by the Reviewer, we also examined Ar protein level in tumors from the Pten knockout model using IHC. Consistent with the previous report (Zhong et al. Cancer Res 74: 1870-1880, 2014), Pten knockout decreased Ar protein level in Pten-null tumors (Fig EV4C). In agreement with the findings in cultured cell lines (Fig 7F and 8A), homozygous deletion of Hdac3 decreased Ar protein levels in Pten-deficient prostate tumors in mice (Fig EV4C), further supporting the notion that HDAC3 likely contributes to PTEN loss-induced tumorigenesis in the prostate by regulating AR protein level.

Minor Comments:

1. For co-IPs, inputs should be blotted for all transfected proteins, particularly if conclusions are being drawn about relative binding (ex. Fig. 4D)

Reply: We apologized for the missing western blot band. We performed new western blot and the new data of Myc-tagged AKT is provided in Fig 4D.

2. IHC and IF should not be described as quantitative

Reply: We thank the Reviewer for the kind suggestion. We have removed the quantitative wording for IHC or IF studies.

3. In figure 2B, it appears that flag-Hdac3 is present in all lanes? In addition the increase in AKT ubiquitination is substantial, yet p-AKT levels are not increased- is this an outlier?

Reply: We repeated the experiment by adjusting the dosage of Flag-HDAC3. New western blot data are shown in Fig 2B.

4. What information is provided in Fig. 2F that is not already provided in Fig. 2D?

Reply: This is an excellent point. We have deleted original Fig 2F.

5. p-AKT blot in Fig. 6c is very over-exposed.

Reply: We agree with the Reviewer. The over-exposed p-AKT blot was replaced by a new western blot data in Fig 6C.

6. why are WT and Hdac3^{-/-} missing from 6c and 6e?

Reply: The reason is that there is no tumor developed in WT or Hdac3^{-/-} group and that deletion of Hdac3 alone had no overt effect on tumorigenesis.

7. review figures for typos (e.g. Fig 6I "fields") and mislabeling (e.g. Fig. 4e- HA-Hdac3 transfected, yet IB says Flag)

Reply: We thank the Reviewer for pointing out these typos and mislabeling. The typos have been corrected throughout the manuscript.

8. Some experiments contain multiple proteins that contain the same epitope tag, which is quite confusing.

Reply: The reason for that is due to the fact that epitope-tagged construct was used for mutagenesis to generate multiple different mutants.

Referee #3 (Comments on Novelty/Model System for Author):

The authors use a combination of genetically modified mouse models, organoids, and PDX as well as cell and molecular biology to address whether HDAC3 affects AKT activity in prostate cancer.

Referee #3 (Remarks for Author):

This is a beautiful piece of research that moves the field forward. I have only a few questions none of which should require additional experimentation. The authors may insert a sentence or two into the paper in response to some of these as they see fit.

Reply: We thank the Reviewer for the very positive comments on our manuscript and kind suggestions regarding how to respond to the comments raised.

p. 6 Figure 1C shows that HDAC3 is the only HDAC required for AKT phosphorylation in C4-2 cells. But this is not necessarily so in all prostate cancer cells. A change in wording please.

Reply: We have changed the wording to "in this cell line" on page 7 as suggested by the Reviewer.

p. 7-8 Figure 2 establishes that HDAC3 and growth factors affect the acetylation and polyubiquitination of AKT. Three forms of Akt, Akt1, 2, and 3 play distinctive roles in signaling and cancer. Do the authors know whether the effects of growth factors and HDAC3 relate to any of particular form of AKT?

Reply: In this study, we used total AKT antibody (cell signaling S9272) that was unable to distinguish three AKT isoforms. Therefore, it is unclear whether or not HDAC3 and growth factors affect the three forms of AKT in the similar manner. We have indicated in METHODS in the revised manuscript that this AKT antibody recognized all three isoforms of AKT.

p. 9 Does the ABD deletion mutant of HDAC3 affect gene expression? Are the domains of HDAC3 important to gene expression and AKT functions common or distinct?

Reply: This is an excellent point. It is generally accepted that the deacetylation enzymatic activity of HDAC3 is important for its role in regulating gene expression. We demonstrated that ABD deletion mutant has similar effect on AKT acetylation as the enzymatic-dead mutant (H134Q) of HDAC3 (Fig 3H). However, whether these two mutants have common or distinct effects on gene expression is unclear at present and warrants further investigation. We have added this discussion in the manuscript.

p. 10 The authors show that in prostate cancer cells cytoplasmic HDAC3 regulates Akt. Another group found that in leukemia, nuclear interactions of these proteins may occur. Do the authors feel that they should indicate that cell or cancer type dependencies may exist? My suspicion is that cytoplasmic interaction may be most important to Akt phosphorylation but why not leave the door open a bit on this issue?

Reply: We agree with the Reviewer. Indeed, we performed new experiments as suggested by Reviewer #2 and demonstrated that the interaction between HDAC3 and AKT also occurred in prostate cancer cells although the interaction was weaker in the nucleus than that in the cytoplasm (Fig. 4B). We have modified our conclusion to reflect this new information.

Can the authors suggest what mediates increased cytoplasmic rather than nuclear HDAC3 activity in prostate cancer with PTEN deletions or SPOP mutations?

Reply: We apologize if our description in the previous version of our manuscript causes any confusion. We have revised our manuscript by emphasizing a few relevant points. In the one hand, given that both AKT and AR signaling are aberrantly activated in SPOP-mutated prostate cancer cells and that both AKT and AR pathways are regulated by HDAC3, our data support the notion that dual inhibition of AKT and AR can be achieved by single targeting of HDAC3 in SPOP-mutated prostate cancer cells. Moreover, because inhibition of AKT activates AR signaling and vice versa, we provide evidence in our manuscript that targeting HDAC3 can inhibit both AKT and AR signaling in PTEN-deficient cells. In the other, we have no evidence suggesting that HDAC3 activity, either in the cytoplasm or in the nucleus, is increased specifically due to PTEN deletion or SPOP mutation in prostate cancer.

p. 16 How significant are the cell type differences in the clonogenic assays (Fig. 7B, C)?

Reply: Clonogenic survival assay demonstrated that C4-2 (PTEN-null/AKT activity high) was more sensitive to GDC0068 compared with 22Rv1 (PTEN-positive/AKT activity low) (Fig 7B). This result is consistent with a previous report that cell lines with a high level of AKT phosphorylation are more sensitive to GDC0068 (Lin, J., et al, Clin Cancer Res 19(7): 1760-72), a notion of 'oncogene addiction'. Interestingly, the steep curves of the clonogenic survival data clearly showed that both cell lines were more sensitive to RGFP966 than GDC0068 (Fig 7B and C). Thus, our data suggest that difference of cell types in terms of PTEN status is important for sensitivity of cells to AKT inhibitor GDC0068. However, cell type difference is less critical for the HDAC3 inhibitor since both cell types are vulnerable to HDAC3 inhibition.

p. 17 Inhibitors of mTOR may have different effects on S6 and EIF 4EBp1. Do the authors have any data or thoughts on the latter mTOR output?

Reply: As suggested by the Reviewer, we performed new western blots to detect the effects of these inhibitors on EIF 4E-BP1 phosphorylation. We demonstrated that both AKT inhibitor GDC0068 and HDAC3 inhibitor RGFP966 affected 4E-BP1 phosphorylation in a trend similar to their effect on S6 phosphorylation, even though the inhibitory effect of GDC0068 on 4E-BP1 phosphorylation was more pronounced than that on S6K phosphorylation in C4-2 cells (Fig 7F).

P18. How effective was the viral transduction of the spheres in figure 8B?

Reply: We performed immunofluorescent cytochemistry (IFC) assay to monitor the viral transduction efficiency using SPOP and HA antibodies. As shown in the Figure below, the lentivirus transfection efficiency was at least 95%.

p. 19 In studies with organoids the authors report a decrease in organoids with a diameter greater than 10 μM . Yet I suspect that there is heterogeneity in the responses of individual organoids to drugs. When spheres are formed in low adherence plates how diverse is the size of the spheres that form? And despite the fact that the organoids (are these really organoids as they come from a cell line?) are genetically homogeneous their response is heterogeneous. Why?

Reply: These are patient-derived organoids, but not from a cell line. We have added a brief description about patient-derived organoids in the METHODS section.

We have to admit that we do not have any genetic/genomic data to clarify whether these organoids are genetically homogeneous or heterogeneous, although they are in very low passage. One plausible explanation for the heterogeneous growth phenotype could be that expression levels of growth factor receptors responsive to Matrigel in some organoids may be altered during the culture/passage of these organoids, although the exact underlying mechanism warrants further investigation.

The authors should describe how they isolated and cultured the patient-derived organoids in more detail.

The authors should explain why they set "30 μm in diameter" as the cut-off value of the growing organoids and how they calculated the percentage of that.

Reply: As indicated above, we have added a new section about patient-derived organoids in METHODS.

In the revised manuscript, we have indicated as follows: Based on the observed growth rate of untreated PTEN-deleted organoids that greater than 50% of organoids reach 30 μm in diameter at day 5. "30 μm " was set as the cut-off value. The number of organoids with the diameter > 30 μm from at least 5 fields were then counted and analyzed

Is HDAC H134Q, a deacetylase inactivation mutant, common in prostate cancer? What is the frequency of this mutation and is there any relationship between prognosis and the frequency of this mutation?

Reply: HDAC3 H134Q is not a prostate cancer-associated mutant. As reported previously by others (Chen, C.S., et al., J Biol Chem 280(46): 38879-87), his mutation causes the loss of deacetylation enzymatic activity of HDAC3. Therefore, this mutant was simply employed as a functional/research tool in our study.

On page 27 of the methods, what is the criterion for establishing the intensity grade of staining? How did the authors divide "low staining" and "strong staining"?

Reply: As we indicated in METHODS in the revised manuscript, staining intensity was graded into four categories: 0, 1, 2 and 3. Specifically, 0 represents no staining, 1 low staining (staining obvious only at X400), 2 medium staining (staining obvious at X100 but not X40), and 3 strong staining (staining obvious at X40).

2nd Editorial Decision

26 January 2018

Thank you for the submission of your revised manuscript to EMBO Molecular Medicine. We have now received the enclosed reports from the referees that were asked to re-assess it. As you will see the reviewers are now globally supportive and I am pleased to inform you that we will be able to accept your manuscript pending a few final amendments:

1) Please address the minor text changes commented by the referees. Please provide a letter INCLUDING my comments and the reviewer's reports and your detailed responses to their comments (as Word file).

***** Reviewer's comments *****

Referee #1 (Remarks for Author):

In response to my earlier comments, authors suggest that it is "clinically practical" to target HDAC3 in tumours. They should cite few latest examples where this is being pursued in appropriate cancer trials.

Regarding the regulation of AR function by HDAC3, authors suggest that the cytoplasmic function of HDAC3 regulates AR activity. Since AR is a transcription factor which is active in the nucleus, the authors should discuss the possible mechanism underlying this regulation.

Referee #2 (Comments on Novelty/Model System for Author):

Beautiful data using in vivo mouse models

Referee #2 (Remarks for Author):

The authors have responded very effectively to the initial round of reviews. The only thing that I would encourage is for the incorporation of SMRT/NCOR into the model. Several different labs have shown that HDAC3 has no deacetylase activity without being bound by the deacetylase activating domain of SMRT or NCOR, perhaps along with an IP4 molecule. While there have been a couple of reports of deacetylase independent functions for HDAC3, the use of inhibitors strongly suggests that this is a deacetylase-dependent function, so SMRT should be included in the model.

2nd Revision - authors' response

7 February 2018

Referee #1 (Remarks for Author):

In response to my earlier comments, authors suggest that it is "clinically practical" to target HDAC3 in tumours. They should cite few latest examples where this is being pursued in appropriate cancer trials.

Reply: We have cited two latest examples where the HDAC3 inhibitor was tested as an anti-cancer agent (Jiang et al., *Cancer Discovery* 7(1): 38-53, 2017; Matthews et al., *Blood* 126(21): 2392-403, 2015). The promising findings in the pre-clinical models might eventually lead to appropriate cancer trials in the future.

Regarding the regulation of AR function by HDAC3, authors suggest that the cytoplasmic function of HDAC3 regulates AR activity. Since AR is a transcription factor which is active in the nucleus, the authors should discuss the possible mechanism underlying this regulation.

Reply: This is an excellent point. We have discussed this in the manuscript (on page 17): "It is worth noting that AR is a transcription factor which is active in the nucleus and it is possible that the cytosolic HDAC3 may modulate AR activity through indirect regulation of AR-regulatory signaling pathway(s)".

Referee #2 (Comments on Novelty/Model System for Author):

Beautiful data using in vivo mouse models

Referee #2 (Remarks for Author):

The authors have responded very effectively to the initial round of reviews. The only thing that I would encourage is for the incorporation of SMRT/NCOR into the model. Several different labs have shown that HDAC3 has no deacetylase activity without being bound by the deacetylase activating domain of SMRT or NCOR, perhaps along with an IP4 molecule. While there have been a couple of reports of deacetylase independent functions for HDAC3, the use of inhibitors strongly suggests that this is a deacetylase-dependent function, so SMRT should be included in the model.

Reply: We thank the Reviewer for the kind suggestion. We have included SMRT as a partner of HDAC3 in the model.

Corresponding Author Name: Runzhi Zhu, Dejie Wang, Haojie Huang

Manuscript Number: EMM-2017-08478